# ORLoopBench: Solver-in-the-Loop Benchmarks for Self-Correction and Behavioral Rationality in Operations Research

**Ruicheng Ao** [1]   **David Simchi-Levi** [1 2 3]   **Xinshang Wang** [4]

## Abstract

Operations Research practitioners debug infeasible models through an iterative process: inspecting Irreducible Infeasible Subsystems (IIS), identifying constraint conflicts, and repairing formulations until feasibility is restored. Existing LLM benchmarks mostly treat OR as one-shot translation from problem descriptions to solver code, omitting this diagnostic loop. We formalize infeasible-model repair as a solver-in-the-loop Markov Decision Process in which each action triggers solver re-execution and IIS recomputation, yielding deterministic, verifiable feedback. We introduce **ORLoopBench**, a benchmark suite with two components: **OR-Debug-Bench** releases 5,362 LP/MILP repair instances, while **OR-Bias-Bench** evaluates closed-form operational decision rationality across inventory settings. Solver-verified RLVR training enables an 8B model to surpass frontier APIs on LP repair (95.3% vs 92.4% RR@5), improves diagnostic behavior, and transfers to MILP repair. The same evaluation exposes semantic drift in whole-model code regeneration: feasible regenerated MILPs can solve the wrong problem. Process-level evaluation with solver oracles enables targeted training for reliable OR self-correction.

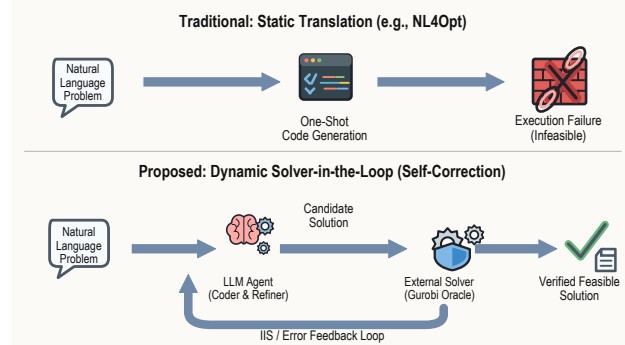

*Figure 1.* Evaluation paradigms compared. **Top**: Static translation benchmarks evaluate one-shot code generation with no execution feedback. **Bottom**: Our solver-in-the-loop approach enables iterative self-correction through IIS feedback.

## 1. Introduction

### 1.1. From Translation to Debugging

When a linear program returns INFEASIBLE, the real work begins. An analyst must examine the Irreducible Infeasible Subsystem (IIS, the minimal subset of constraints that cannot be simultaneously satisfied), diagnose the root cause, and systematically repair the formulation. This iterative debugging loop is where OR expertise manifests.

Yet existing benchmarks evaluate LLMs on Operations Research as one-shot translation: given a problem description, generate solver code. This paradigm ignores the debugging process central to real OR practice. Unlike generic error messages, IIS provides a **minimal certificate of infeasibility**, enabling targeted, interpretable repairs. This structured feedback is what should enable targeted self-correction.

### 1.2. Self-Correction in Structured Domains

Recent work showed that 64.5% of LLM errors result when models fail to self-correct (Tie et al., 2025). However, CorrectBench focuses on general programming tasks and omits the OR domain, where structured self-correction is uniquely suited. First, solvers provide **deterministic feedback**: precise, verifiable signals such as IIS, slack values, and objective bounds. Second, **verifiable ground truth** enables

---

[1]Institute for Data, Systems, and Society, Massachusetts Institute of Technology, Cambridge, MA, USA [2]Operations Research Center, Massachusetts Institute of Technology, Cambridge, MA, USA [3]Department of Civil and Environmental Engineering, Massachusetts Institute of Technology, Cambridge, MA, USA [4]Alibaba Group. Correspondence to: Ruicheng Ao <aorc@mit.edu>, David Simchi-Levi <dslevi@mit.edu>, Xinshang Wang <xinshang.w@alibaba-inc.com>.

*Proceedings of the $43^{rd}$ International Conference on Machine Learning*, Seoul, South Korea. PMLR 306, 2026. Copyright 2026 by the author(s).

mathematical checking of optimal solutions. Third, the **interpretable process** means the diagnostic reasoning chain admits structured evaluation.

This combination (deterministic oracle, verifiable outcomes, structured process) makes OR a natural testbed for studying self-correction. The solver-in-the-loop paradigm forces agents to reason from IIS feedback, enabling systematic hypothesis refinement rather than blind trial-and-error.

### 1.3. Behavioral Rationality in Operations

While debugging addresses *upstream* formulation errors, operational decisions face a distinct *downstream* challenge. Concurrent work on AIM-Bench (Zhao et al., 2025) revealed systematic behavioral biases in LLM inventory managers: a "pull-to-center" tendency where models over-order when optimal quantity is low and under-order when it is high. This bias persists across model scales, raising concerns about deploying LLMs in high-stakes operations management.

### 1.4. Contributions

We make four contributions:

1. **A solver-in-the-loop MDP for OR debugging**: We formalize infeasible-model repair as a sequential decision problem in which each action modifies the formulation, triggers solver re-execution, and receives updated IIS feedback. This shifts evaluation from static NL-to-code translation to the diagnostic loop used in practice.

2. **Solver-verified training for small models**: We adapt RLVR to OR repair with rewards for feasibility recovery, objective preservation, diagnostic accuracy, and faithful use of IIS evidence. An 8B model trained with GRPO surpasses the strongest frontier API in LP repair: 95.3% vs 92.4% RR@5. In the core 26-model evaluation, it also improves DA by +14.6 pp.

3. **ORLoopBench**: ORLoopBench consists of two controlled components. OR-Debug-Bench releases 5,362 LP/MILP repair instances spanning LP error types A–I and MILP repair settings. OR-Bias-Bench evaluates operational decision rationality in newsvendor and EOQ inventory settings with ID/OOD splits. The benchmark files are available at https://github.com/Archer222arc/ORLoopBench.

4. **Benchmark findings**: OR-Debug-Bench compares against OR-domain baselines (ORLM, SIRL, LL-MOPT, OptiMUS) and includes semantic-drift analysis for code regeneration; OR-Bias-Bench includes one-shot and feedback-based decision protocols. The MDP repair framework transfers to MILP, reaching 87.1% RR@5 versus 71.0% for the best API baseline.

*Table 1.* Key results summary.

| Finding | Metric | Result |
|---|---|---|
| 8B surpasses frontier APIs | RR@5 | 95.3% vs 92.4% |
| Diagnostic accuracy gain | DA | 62.4% vs 47.8% |
| Efficiency improvement | Steps | 2.25 vs 3.15 |
| OOD generalization | Bias drift | -9.6% (best OOD) |
| Bias reduction | Bias diff | 20.0%→10.4% |

*Table 2.* ORLoopBench benchmark positioning. OR-Debug-Bench evaluates iterative debugging with deterministic oracle feedback.

| Benchmark | Year | Task | Oracle | Multi-step | Self-Corr. |
|---|---|---|---|---|---|
| NL4Opt (Ramamonjison et al., 2023) | 2022 | NL→LP | – | – | – |
| OptiBench (Yang et al., 2024) | 2024 | Formulation | – | – | – |
| MAMO (Huang et al., 2025b) | 2024 | Complex LP | – | – | – |
| ORLM (Huang et al., 2025a) | 2025 | Formulation | – | – | – |
| AIM-Bench (Zhao et al., 2025) | 2025 | Inventory | Closed-form | – | – |
| SWE-bench (Jimenez et al., 2024) | 2024 | Code Debug | Unit Tests | ✓ | Limited |
| CorrectBench (Tie et al., 2025) | 2025 | Self-Corr. | – | – | General |
| **OR-Debug-Bench** | **2026** | **Debugging** | **Gurobi IIS** | **✓** | **✓** |
| **OR-Bias-Bench** | **2026** | **Decision** | **Closed-form** | **–** | **✓** |

## 2. Benchmark Setup

### 2.1. OR-Debug-Bench: Data Organization & Metrics

ORLoopBench organizes our benchmark suite into two components: OR-Debug-Bench for upstream solver-guided model repair and OR-Bias-Bench for downstream closed-form operational decisions.

OR-Debug-Bench turns infeasibility repair into an interactive task rather than a one-shot code-generation task. Each instance provides a natural-language problem description and sabotaged Gurobi code that returns Infeasible. The hidden ground truth contains the original feasible code, the sabotaged constraint, and the intended repair. During evaluation, Gurobi 11.0 computes an IIS after each attempted repair, giving the agent a deterministic certificate of the current conflict. Figure 3 illustrates a complete episode where the agent diagnoses an IIS conflict and repairs the model in two steps.

**Data Organization.** The released OR-Debug-Bench file contains 5,362 LP/MILP repair instances; the JSON field names in the public repository are storage labels, not separate benchmark definitions. Training uses 996 generation artifacts produced during benchmark construction. The LP repair evaluation uses 450 instances (50 per error type A–I), and the MILP repair evaluation uses repeated runs across 10 domains and 8 error types. For OR-Bias-Bench, training uses 900 samples across three curriculum stages with CR range [0.3, 0.7], while evaluation covers broader ranges: ID [0.05, 0.95] and OOD [0.10, 0.89].

**Metrics.** We report three complementary quantities. RR@k measures the percentage of instances repaired to Optimal within $k$ repair steps in a single solver-interaction episode. DA measures whether the agent identified the true conflict-

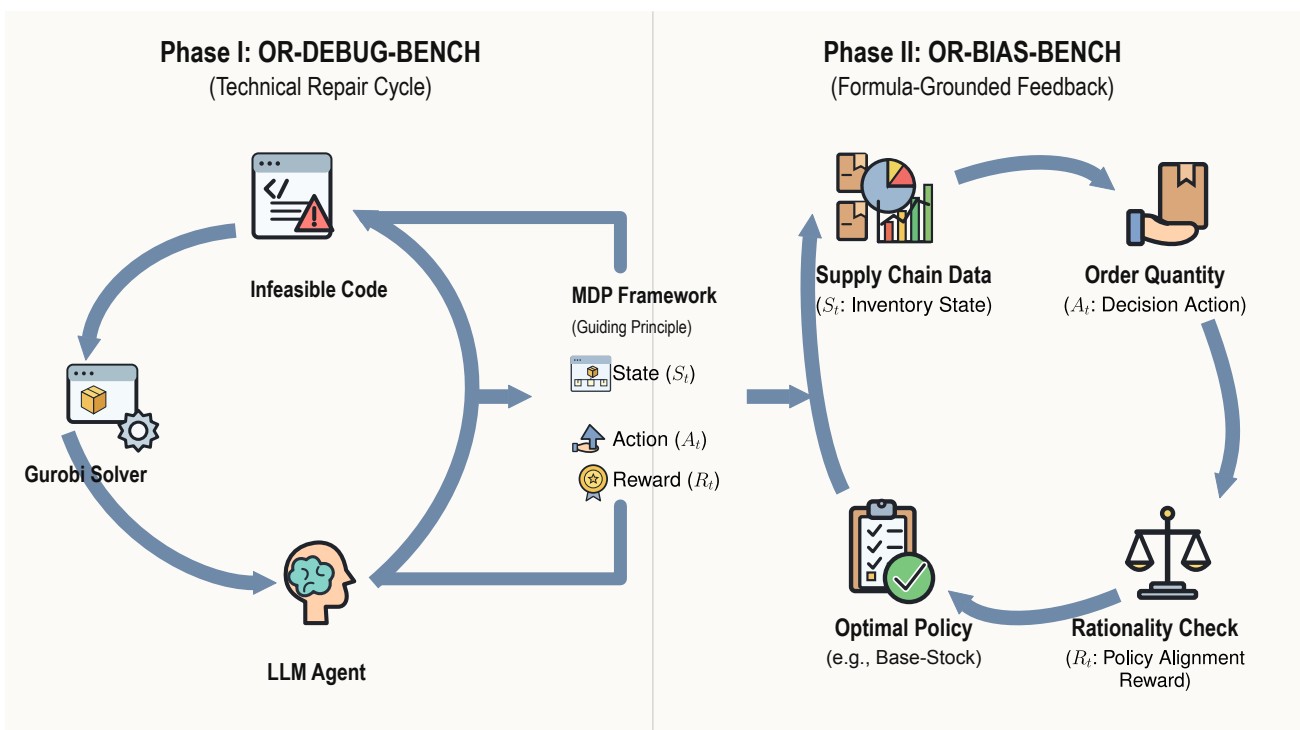

*Figure 2.* Two-phase benchmark framework. **Phase I (OR-DEBUG-BENCH)**: Iterative debugging where the agent receives Gurobi IIS feedback and repairs infeasible code. **Phase II (OR-BIAS-BENCH)**: Inventory decision-making verified against closed-form optimal policies.

*Table 3.* OR-DEBUG-BENCH benchmark statistics and evaluation protocols.

| Attribute | Value |
|---|---|
| *Dataset Structure* | |
| Released Instances | 5,362 |
| Scope | LP/MILP infeasibility repair |
| LP Error Types | 9 (A–I) |
| MILP Error Types | 8 |
| IIS Range | 1–11 constraints |
| Difficulty Levels | Easy / Medium / Hard |
| *Reported Protocols* | |
| LP Repair | 450 instances (50 per error type) |
| MILP Repair | 10 domains × 8 error types, 5 repeats |
| *Training Artifacts* | |
| Training (SFT) | 696 trajectories |
| Training (RL) | 300 prompts |
| Max Steps per Episode | 50 |

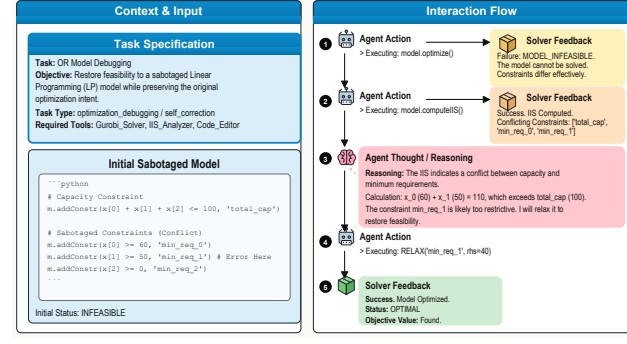

*Figure 3.* Example OR-DEBUG-BENCH episode. **Left**: The agent receives a sabotaged LP where minimum requirements ($60 + 50 + 0 = 110$) exceed capacity (100). **Right**: The agent (1) attempts optimization, (2) computes IIS, (3) reasons about the conflict, (4) relaxes the key constraint, and (5) achieves OPTIMAL status in 2 repair steps.

ing constraints, rather than reaching feasibility by accident. OP measures whether the repaired model preserves the original objective value.

### 2.2. OR-BIAS-BENCH: Data Organization & Metrics

OR-BIAS-BENCH evaluates whether models make operational decisions that agree with closed-form optima. It contains 2,000 newsvendor instances (1,000 ID + 1,000 OOD), where the optimal order quantity is $Q^* = F^{-1}(\text{CR})$, and 300 EOQ instances, where $Q^* = \sqrt{2DK/h}$. The same evaluation framework supports both one-shot decisions and a multi-turn feedback protocol in which the model may revise its quantity after seeing formula-grounded cost feedback.

**Metrics.** Rationality measures valid response percentage.

*Table 4.* OR-BIAS-BENCH data splits used in experiments. The benchmark contains 2,000 instances (1,000 ID + 1,000 OOD); we evaluate on stratified subsets.

| Attribute | Train | ID Eval | OOD Eval |
|---|---|---|---|
| Samples | 900 | 400 | 200 |
| CR Range | [0.3, 0.7] | [0.05, 0.95] | [0.10, 0.89] |
| Demand Dist. | $\mathcal{N}(\mu, \sigma)$ | $\mathcal{N}(\mu, \sigma)$ | $\mathcal{N}(\mu, \sigma)$ |
| $\mu$ Range | [50, 200] | [50, 200] | [50, 200] |
| $\sigma$ Range | [10, 50] | [10, 50] | [10, 50] |

*Table 5.* ORLOOPBENCH components. OR-DEBUG-BENCH targets upstream model repair through iterative solver interaction, while OR-BIAS-BENCH evaluates downstream decision-making against analytical ground truth.

| Aspect | OR-DEBUG-BENCH | OR-BIAS-BENCH |
|---|---|---|
| Domain | Mathematical Programming | Operations Management |
| Task | Debug infeasible code | Inventory decision |
| Oracle | Gurobi IIS feedback | Closed-form $Q^* = F^{-1}$(CR) |
| Interaction | Multi-step solver episode | Single-shot + feedback diagnostic |
| LLM Challenge | Error parsing, code repair | Cognitive bias mitigation |
| Key Metrics | RR@k, DA, OP | Rationality, Bias Diff |

Bias Diff = $|\mathbb{E}[Q/Q^*|\text{CR} > 0.5] - \mathbb{E}[Q/Q^*|\text{CR} < 0.5]|$ measures pull-to-center. ID→OOD $\Delta$ measures generalization.

### 2.3. Evaluation Protocol

For OR-DEBUG-BENCH, an episode alternates between model actions and solver feedback: the agent sees the current infeasible model and IIS, edits or queries the formulation, and receives the updated solver status. The loop stops at OPTIMAL or the step budget. For OR-BIAS-BENCH, the agent outputs an order quantity $Q$ and, in the feedback protocol, may revise it after receiving closed-form cost feedback.

We evaluate 26 models on the LP OR-DEBUG-BENCH repair task and report MILP repair results on random instances with repeated runs. OR-BIAS-BENCH results cover the newsvendor and EOQ inventory settings across frontier APIs and local variants.

## 3. Benchmark Construction

### 3.1. Saboteur-based Problem Generation

We design a "saboteur" pipeline that injects controlled errors into valid linear programs while maintaining verifiable ground truth. Each generated error must: (1) produce a verifiable INFEASIBLE status, (2) yield a non-empty IIS containing the sabotaged constraint, and (3) have a unique ground-truth fix restoring OPTIMAL status.

**Generation Pipeline.** The pipeline operates in four stages: (1) source selection from a feasible LP pool, (2) sabotage application using type-specific corruption, (3) verification

*Table 6.* Error type taxonomy for OR-DEBUG-BENCH.

| Type | Name | Difficulty |
|---|---|---|
| A | Direction Flip | Hard |
| B | Variable Type Error | Easy |
| C | Coefficient Modification | Easy |
| D | Contradicting Constraint | Hard |
| E | Multi-Constraint Conflict | Hard |
| F | Hidden Dependency | Hard |
| G | Cascading Conflict | Hard |
| H | IIS-Incomplete | Medium |
| I | Optimal Selection | Medium |

*Table 7.* Difficulty calibration for OR-DEBUG-BENCH. Levels are defined by baseline API model performance.

| Level | Error Types | Baseline RR@5 |
|---|---|---|
| Easy | B, C | $\geq 85\%$ |
| Medium | H, I | 70–85% |
| Hard | A, D, E, F, G | $< 70\%$ |

via Gurobi IIS computation, and (4) oracle labeling for evaluation. Of the initial candidate pool, 87% pass all validation checks on first generation; the remainder require at most two iterations. Full algorithmic details appear in Appendix D.1.

**Robust Injection.** Reliable error injection uses adaptive methods: Type A (direction flip) uses slack-based constraint selection to improve success from 30% to 95%; Type C (upper bound conflict) uses a 4-tier fallback strategy achieving 72% success. Complete algorithms appear in Appendix D.8.

### 3.2. Anti-Pattern Measures

Three mechanisms prevent pattern-matching shortcuts. First, **randomized naming** in Types G–I uses UUID-based identifiers to prevent models from exploiting semantic name patterns. Second, **hidden dependencies** (Type F) create scenarios where the IIS reveals a symptom constraint while the root cause lies elsewhere. Third, **cascading conflicts** (Type G) require multi-step reasoning: fixing the primary conflict reveals a secondary one. Appendix D.10 provides the construction details.

### 3.3. Newsvendor Problem Generation

For OR-BIAS-BENCH, we generate newsvendor scenarios with controlled critical ratios:

$$Q^* = \mu + \sigma \cdot \Phi^{-1}(\text{CR}), \quad \text{CR} = \frac{p - c}{p - s} \quad (1)$$

where $\Phi^{-1}$ is the standard normal inverse CDF, $\mu$ is mean demand, and $\sigma$ is standard deviation.

**Stratified Sampling.** We stratify by CR buckets: ID covers [0.05, 0.95] with 100+ samples per bucket; OOD covers [0.10, 0.89] testing intermediate-to-extreme values.

*Table 8.* Evaluation difficulty levels for OR-BIAS-BENCH. Each level targets specific bias phenomena with controlled CR ranges and prompt complexity.

| Level | CR Range | Prompt Style | Target Phenomenon |
|---|---|---|---|
| L1 | $[0.4, 0.6]$ | Clean | Foundation (neutral) |
| L2 | $[0.05, 0.2] \cup (0.8, 0.95]$ | Clean | Bias trigger (extreme) |
| L3 | $[0.3, 0.7]$ | + Distractors | Robustness test |
| L4 | $[0.1, 0.9]$ | + Censored | Expert inference |

The four evaluation difficulty levels are detailed in Appendix D.3.

### 3.4. Interaction Model

The central methodological step is to make solver feedback part of the state transition. Instead of asking a model to regenerate code from scratch, OR-DEBUG-BENCH exposes a small set of repair actions and reruns the solver after every action. The next state is therefore not sampled or judged heuristically; it is the deterministic result of Gurobi execution and IIS recomputation.

**OR-DEBUG-BENCH state and actions.** The state contains the natural-language problem, current code, solver status, current IIS, action history, and step index. Actions fall into three groups: diagnostic queries such as GET_IIS and CHECK_SLACK; repair actions such as RELAX, DROP, and REWRITE; and SUBMIT for episode termination. Full state/action specifications appear in Appendix E.1.

**Reward.** The reward combines three goals:

$$\mathcal{R} = 0.5 R_{\text{outcome}} + 0.3 R_{\text{diagnosis}} + 0.2 R_{\text{efficiency}}. \quad (2)$$

Outcome rewards check whether the repaired model reaches OPTIMAL; diagnostic rewards check whether the agent identified the ground-truth conflicting constraints; efficiency rewards favor shorter repair sequences. This design makes success, explanation quality, and repair cost visible separately.

**OR-BIAS-BENCH interaction.** The bias task is single-step in its one-shot form: the model chooses an order quantity and the oracle compares it with the closed-form optimum in Eq. (1). The feedback variant repeats this decision after reporting the realized cost and optimal cost.

**Diagnostic Accuracy (DA).** We measure alignment between diagnosed constraints and ground truth:

$$\text{DA} = \frac{|\text{diagnosed} \cap \text{IIS}_{\text{GT}}|}{|\text{IIS}_{\text{GT}}|} \quad (3)$$

DA measures coverage of the true conflict set; false-positive diagnoses and off-target edits are tracked separately through precision-style diagnosis checks, the faithfulness penalty, and OP. High RR@5 with low DA indicates "lucky" solutions that fix problems without understanding root causes.

*Table 9.* Pilot study: foundation model screening on OR-DEBUG-BENCH validation.

| Model | Base RR@5 | +SFT RR@5 | $\Delta$ |
|---|---|---|---|
| **Qwen3-8B** | **51.2%** | **93.1%** | **+41.9%** |

## 4. Training Methods

Figure 4 illustrates the two training tracks. Both start from Qwen3-8B. The debugging track first teaches the model the action format and basic IIS interpretation through supervised fine-tuning, then uses solver-verified reinforcement learning to optimize repair success. The bias track uses supervised examples followed by a curriculum designed to counter pull-to-center behavior.

### 4.1. Foundation Model Selection

We selected Qwen3-8B-Instruct based on a pilot study (100 samples) showing +41.9% post-SFT improvement headroom.

Qwen3-8B achieved 93.1% RR@5 after SFT with efficient token usage (2,100 tokens/episode). See Appendix J for methodology.

### 4.2. Supervised Trajectories

We collect successful debugging trajectories from three teacher models (GPT-5.2-chat 40%, o4-mini 35%, DeepSeek-R1 25%), chosen for diverse reasoning styles. We retain trajectories that solve the instance within five steps and diagnose at least half of the ground-truth conflict. This filtering yields 696 of 1,247 trajectories (55.8% acceptance), averaging 2.3 steps with 68% diagnostic accuracy. For OR-BIAS-BENCH, we collect 500 rational responses. Training examples are shown in Appendix E.10.

### 4.3. GRPO Training with Composite Reward

Following DeepSeek-R1 (DeepSeek-AI, 2025), we use Group Relative Policy Optimization with KL removal ($\beta = 0$), asymmetric clipping $[0.2, 0.28]$, and LoRA ($r = 16$, $\alpha = 32$).

**Composite Reward.** The reward balances outcome verification, diagnostic quality, and efficiency:

$$R = 0.5 \cdot R_{\text{outcome}} + 0.3 \cdot R_{\text{diagnosis}} + 0.2 \cdot R_{\text{efficiency}} \quad (4)$$

where $R_{\text{outcome}} = +100$ for OPTIMAL and $-50$ otherwise, $R_{\text{diagnosis}} = \text{DA} \cdot 100$, and $R_{\text{efficiency}} = -1$ per step. The diagnostic term addresses trial-and-error repairs that restore feasibility without identifying the root cause. A faithfulness penalty ($-20$) discourages repairs targeting non-IIS constraints; without this penalty, models often achieve OPTIMAL through indirect fixes that mask the root cause. Train-

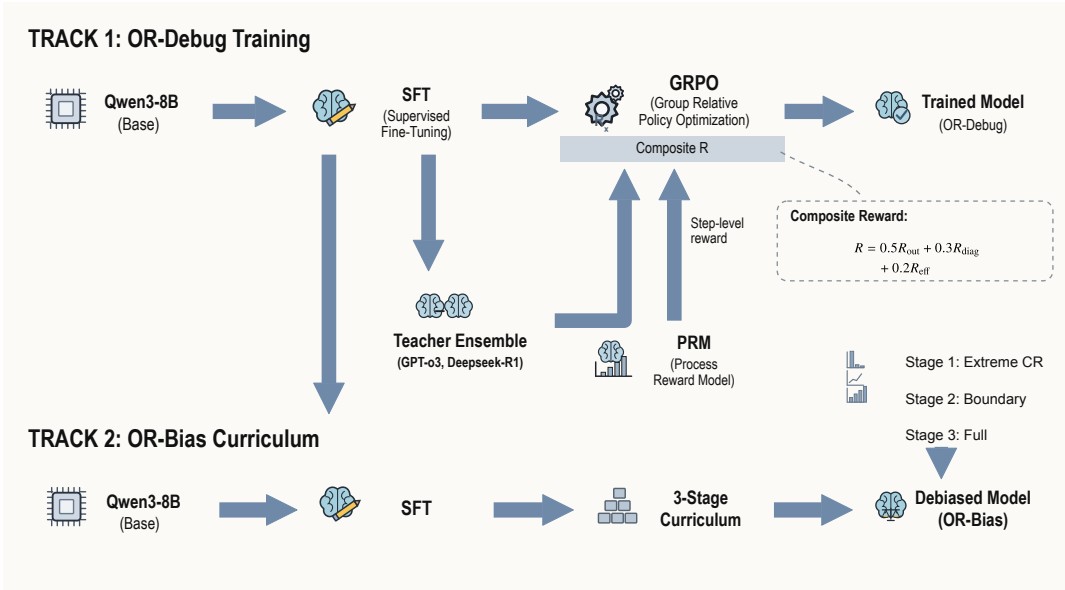

*Figure 4.* Training pipeline overview. Track 1 trains the OR-DEBUG-BENCH model: SFT on teacher trajectories followed by GRPO with composite reward and optional PRM supervision. Track 2 trains the OR-BIAS-BENCH model: SFT on rational responses followed by a three-stage curriculum (Extreme → Boundary → Full) that targets pull-to-center bias.

*Table 10.* Three-stage curriculum for OR-BIAS-BENCH.

| Stage | Focus | Samples | CR Distribution |
|---|---|---|---|
| 1 | Direction Learning | 200 | Extreme (0.1, 0.9) |
| 2 | Boundary Refinement | 300 | Near-boundary |
| 3 | Full Distribution | 400 | [0.2, 0.8] |

ing converges after 4 epochs with RR@5=95.0%. Full training curves appear in Appendix E.5.

### 4.4. Process Supervision

Outcome-based rewards provide sparse feedback: many wrong actions only reveal their cost after several solver calls. We therefore train a process reward model (PRM) to score individual steps. A step receives the highest label if it reaches OPTIMAL or shrinks the IIS, a medium label if it identifies a ground-truth conflict, a small label for useful diagnostic actions, and zero otherwise. The PRM achieves AUC-ROC of 0.94 on held-out labels and improves DA by +4.7 pp over SFT (68.0%→72.7%), while the curriculum variant achieves higher RR@5. Details appear in Appendix E.4.

### 4.5. Curriculum Learning for Bias Mitigation

For OR-BIAS-BENCH, we use a three-stage curriculum targeting the pull-to-center bias:

**Stage 1 (Direction Learning)** uses extreme CR values (0.1, 0.9) to teach whether quantity should move up or down.

**Stage 2 (Boundary Refinement)** uses near-boundary values ([0.15, 0.25] and [0.75, 0.85]) to refine magnitude estimation. **Stage 3 (Full Distribution)** covers [0.2, 0.8] to consolidate learning.

This staged approach achieves 48% bias reduction (20.0%→10.4%) on OOD scenarios. Among the trained local variants, curriculum training is the only approach with improved OOD bias relative to ID (−9.6% drift). Analysis appears in Appendix E.6.

## 5. Experiments

### 5.1. Experimental Setup

We evaluate 26 models: 4 local QWEN3-8B variants (SFT, GRPO, DAPO, Curriculum) and 22 API models spanning Claude, GPT, o-series, DeepSeek, Gemini, Qwen API, Llama, and kimi-k2. Experiments run on 2×A100 80GB with SGLang inference (TP=2, concurrency=16).

### 5.2. Main Results

Table 11 reports the core 26-model evaluation, including diagnostic accuracy and step efficiency. QWEN3-8B-GRPO reaches 95.3% RR@5, 62.4% DA, and 2.25 steps on average. Against the frontier summary in Appendix F.4, the strongest LP API is Claude Sonnet 4.6 at 92.4% RR@5, so QWEN3-8B-GRPO remains ahead by +2.9 pp. In the core evaluation, the trained model also improves DA by +14.6 pp and uses fewer repair steps than o4-mini, the step-efficient top-RR@5 core API baseline (2.25 vs 3.15).

*Table 11.* OR-DEBUG-BENCH LP results on representative models. The test set contains 450 instances, selected as 50 held-out problems from each error type A–I. Full results for all 26 models appear in Appendix F.

| Model | RR | RR@5 | DA | Steps |
|---|---|---|---|---|
| **QWEN3-8B-GRPO** | 100% | **95.3%** | **62.4%** | **2.25** |
| QWEN3-8B-Curriculum | 100% | 94.0% | 61.7% | 2.22 |
| QWEN3-8B-DAPO | 100% | 93.8% | 60.4% | 2.31 |
| QWEN3-8B-SFT | 99.8% | 93.1% | 60.8% | 2.34 |
| o4-mini | 97.8% | **86.2%** | 47.8% | **3.15** |
| claude-sonnet-4 | **100%** | 86.2% | 50.1% | 3.71 |
| o1 | 99.8% | 82.9% | 47.8% | 3.78 |
| gpt-5.2-chat | 99.8% | 81.8% | 40.9% | 3.72 |
| gemini-2.5-flash | 84.2% | 70.7% | 19.2% | 3.23 |
| Llama-3.3-70B | 93.8% | 60.9% | 46.9% | 4.81 |
| DeepSeek-V3.2 | 99.3% | 58.9% | 44.8% | 4.86 |
| DeepSeek-R1 | 99.1% | 56.7% | 34.5% | 5.08 |

The trained models use a "diagnose once, repair correctly" pattern: 1.3 diagnostic actions per episode vs 2.1 for API models, then targeted repairs. This reflects a different reasoning strategy: systematic elimination rather than trial-and-error.

**Per-Error-Type Performance.** Domain-specific training provides larger gains on harder problems (A, D–G): +9.6% average (94.4% vs 84.8%). Easy types (B, C) show smaller gains (+3.0%) as baselines already exceed 95%. Medium types (H, I) improve by +14.0% (95.0% vs 81.0%). Full breakdown appears in Appendix F.2.

**Cost-Performance Trade-off.** Local deployment avoids per-call API charges in our setup. Training cost ($\sim$8 GPU-hours on 2$\times$A100) amortizes across high-volume evaluation, and production cost depends on utilization and infrastructure.

## 5.3. MILP and OR-Domain Baselines

OR-domain baselines show that solver-in-the-loop repair is not solved by existing OR formulation models. We evaluate ORLM, SIRL, and LLMOPT in the same MDP environment with identical IIS observations, action space, system prompt, and step budget. OptiMUS is evaluated in its native code-regeneration mode because it does not produce constraint-level repair actions. On LP repair, ORLM reaches 71.0% RR@5, SIRL 73.8%, LLMOPT 78.4%, and OptiMUS 28.9%, compared with 95.3% for our trained repair model.

The same pattern holds on MILP. Using random MILP evaluation instances across 10 domains and 8 error types with five repeated runs, the LP-trained model transfers zero-shot at 78.8% RR@5; MILP-specific training reaches 87.1%. Baselines degrade more sharply: the best API baseline, Claude Sonnet 4.6, reaches 71.0%, SIRL reaches 67.6%, LLMOPT

*Table 12.* OR-BIAS-BENCH results (400 ID / 200 OOD samples). Bias = difference from rational ordering.

| Model | ID Bias | OOD Bias | Δ | Status |
|---|---|---|---|---|
| claude-haiku-4.5 | **0.0%** | 3.6% | +3.6% | Best ID |
| QWEN3-8B-OM-SFT | 4.9% | 11.5% | +6.6% | OK |
| o4-mini | 6.7% | 7.7% | +1.0% | OK |
| **QWEN3-8B-Curriculum** | 20.0% | **10.4%** | -9.6% | **Best OOD** |
| gpt-4.1 | 11.8% | 15.4% | +3.6% | OK |
| Llama-3.3-70B | 19.2% | 12.5% | -6.7% | OK |
| gpt-5-mini | 1.2% | 53.3% | +52.1% | Degraded |

44.4%, and OptiMUS 2.2%. Thus, mixed-integer structure reduces absolute performance, but the MDP repair interface and solver-verified training remain effective without architectural changes.

**Semantic drift in code regeneration.** OptiMUS-style regeneration can produce executable and feasible models that no longer encode the intended problem. On the MILP semantic-drift evaluation, GPT-5.4 reaches OPTIMAL solver status in 90% of cases, but only 28.2% preserve the correct objective; Claude Sonnet 4.6 reaches 85% OPTIMAL but 22.4% correct objective, and Gemini 3.1 Pro reaches 3% OPTIMAL and 0.8% correct objective. Constraint-level repair avoids this failure mode by preserving the objective and editing only targeted constraints.

## 5.4. Bias Evaluation Results

Curriculum training achieves the best OOD generalization with $-$**9.6%** drift (20.0%$\rightarrow$10.4%), the only trained model with substantial OOD improvement. Several API models show increased OOD bias; gpt-5-mini degrades sharply from 1.2% to 53.3%, while o4-mini remains comparatively stable (+1.0%). While claude-haiku-4.5 achieves lowest ID bias (0%), curriculum training shows *improved* OOD performance ($-$9.6% drift vs $-$6.7% for Llama-3.3-70B).

OR-BIAS-BENCH includes both newsvendor and EOQ inventory decisions. On the 300-instance EOQ setting and a five-round multi-turn protocol with closed-form error feedback, DeepSeek-R1 reduces bias from 56.9% to 0.5% in 2.6 rounds on average, while GPT-5.2 corrects immediately; other models remain problem-dependent, indicating that solver- or formula-grounded feedback improves some but not all operational decision biases.

## 5.5. Inference Scaling Analysis

Figure 5 shows RR@k scaling. QWEN3-8B-GRPO at $k = 3$ (92.1%) already surpasses o4-mini at $k = 10$ (90.7%). Token efficiency is 2.8$\times$ better: 2,109 tokens per success vs 5,976 for o4-mini. Hard problems show steeper scaling: +26.8% from $k = 1$ to $k = 5$ vs +9.8% for Easy problems. Full analysis appears in Appendix I.

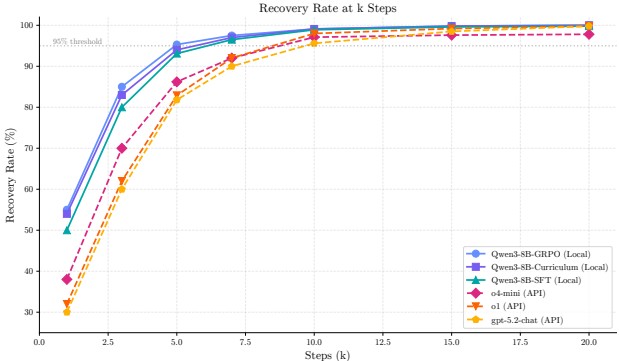

*Figure 5.* Recovery Rate vs. repair-step budget $k$ in the core 26-model evaluation. QWEN3-8B-GRPO reaches 95.3% RR@5; Appendix F.4 reports the frontier API summary.

*Table 13.* Ablation study on OR-DEBUG-BENCH (200-sample validation set).

| Configuration | RR@5 | DA | ΔRR@5 |
|---|---|---|---|
| SFT Baseline | 91.5% | 68.0% | – |
| + GRPO | 92.0% | 66.0% | +0.5% |
| + Curriculum | 95.0% | 68.0% | +3.5% |
| + PRM | 92.0% | 72.7% | +0.5% |
| Curriculum + GRPO (best) | **95.3%** | 62.4% | **+3.8%** |

### 5.6. Ablation Study

Three ablation results stand out: (1) Curriculum pre-training provides +3.5% RR@5, the largest single improvement; (2) relative to SFT, PRM raises DA from 68.0% to 72.7% with similar RR@5 (91.5% to 92.0%), while relative to curriculum it prioritizes DA over RR@5; (3) Curriculum synergizes with GRPO (+3.8% over SFT), providing favorable initialization for hard problems.

## 6. Related Work

**OR Benchmarks and Formulation Systems.** Table 2 compares ORLOOPBENCH with existing OR benchmarks. NL4Opt (Ramamonjison et al., 2023) initiated natural-language-to-LP evaluation; later work expanded prompt-based and learning-based formulation systems (Xiao et al., 2024; AhmadiTeshnizi et al., 2024; Yang et al., 2024; Huang et al., 2025a; Chen et al., 2025b; Liu et al., 2026). Related benchmarks and agents cover OR reasoning, LP formulation, workflow execution, and dynamic programming (Jiang et al., 2024; Huang et al., 2025b; Mostajabdaveh et al., 2025; Ao et al., 2026e; Zhou et al., 2025). These efforts primarily evaluate static, one-shot formulation. OR-DEBUG-BENCH instead starts from an already-built infeasible model, exposes live IIS feedback after each repair, and evaluates iterative constraint-level correction under objective-preservation checks. A broader comparison appears in Appendix A.

**Formulation Training vs Debugging.** Recent systems use solvers, search, or localized data construction to improve formulation quality (Chen et al., 2025b; Astorga et al., 2025; Liu et al., 2025; 2026). OptiChat studies natural-language interaction with optimization models, including infeasibility diagnosis through solver tools (Chen et al., 2024; 2025a). A supply-chain-focused predecessor, OptiRepair (Ao et al., 2026f), studies closed-loop diagnosis and repair for multi-echelon inventory models. These works are close in spirit but differ in scope: OR-DEBUG-BENCH provides a controlled benchmark, MDP action interface, objective-preservation checks, and solver-verified training/evaluation for iterative constraint-level repair. This distinction matters empirically: formulation-trained models trail repair-trained models under the same MDP interface, and whole-model regeneration can solve to OPTIMAL while changing the intended objective or constraints.

**Self-Correction and Verifiable Rewards.** General self-correction work studies iterative refinement, reasoning bootstrapping, code debugging, and RL-trained correction (Madaan et al., 2023; Zelikman et al., 2022; Jimenez et al., 2024; Tie et al., 2025; Kumar et al., 2024). These settings typically rely on self-generated feedback or sampled tests. In contrast, solver feedback is deterministic: IIS computation identifies infeasibility certificates and supports automatically checkable progress signals. Our GRPO and PRM training build on RLVR and process-supervision methods (Schulman et al., 2017; Ouyang et al., 2022; Rafailov et al., 2023; Shao et al., 2024; DeepSeek-AI, 2025; Lightman et al., 2024; Wang et al., 2024; Zhang et al., 2025a; Setlur et al., 2025), but adapt them to OR debugging where each repair step can be verified by the solver.

**Behavioral Rationality in LLM Decisions.** Recent OM work examines LLMs and predictive models as decision inputs in demand simulation, pricing, service-system selection, and inventory decisions (Zhang et al., 2025b; Ao et al., 2026c;a; Zhao et al., 2025). OR-BIAS-BENCH builds on the pull-to-center phenomenon documented in LLM inventory choices and classical human newsvendor bias (Schweitzer & Cachon, 2000; Kremer et al., 2010). It adds explicit ID/OOD splits and formula-grounded feedback protocols, showing that curriculum learning can reduce OOD decision bias rather than only improve in-distribution fit.

## 7. Discussion

ORLOOPBENCH is organized around two complementary failure modes. OR-DEBUG-BENCH targets upstream formulation repair: models must act on Gurobi IIS feedback and reason about constraint conflicts, where API models rely more on trial-and-error. OR-BIAS-BENCH targets downstream decision quality: models must produce closed-form inventory decisions that remain rational under ID/OOD

shifts, where gpt-5-mini's 1.2% ID bias and 53.3% OOD bias reveal brittle heuristics. The training response differs accordingly: GRPO improves OR-DEBUG-BENCH through solver-verified rewards, while curriculum training reduces OR-BIAS-BENCH OOD bias by staging exposure to extreme CR values.

Three design choices prevent shortcut solutions. OP penalizes feasible repairs that alter the objective; the faithfulness penalty discourages edits outside the current IIS; and DA separates root-cause diagnosis from lucky feasibility restoration. These checks matter most in MILP code regeneration, where a model can produce executable code that solves to OPTIMAL while changing objective coefficients or constraint semantics. Constraint-level repair narrows the action space and preserves the original objective.

Standard prompting remains insufficient because OR debugging requires procedural knowledge: iterative solver interaction, domain-specific diagnostics, and state-dependent repair strategy. Zero-Shot CoT reaches 23.0% RR@5; three-shot ICL reaches 38.7%, still 54 points below SFT. Appendix J.2 analyzes these prompting baselines.

The remaining failures point to the limits of the current interface. Type H–I failures often involve large IIS sets where one repair exposes another conflict; false positives occur when constraints play similar semantic roles; and repair-magnitude errors suggest hybrid approaches in which the model identifies the faulty constraint and an optimizer computes the smallest valid adjustment.

**Limitations.** OR-DEBUG-BENCH covers infeasible LP/MILP repair with Gurobi-verifiable feedback; nonlinear, stochastic, robust, and multi-objective formulations require different certificates. IIS is a minimal infeasible subset rather than a complete causal explanation, and multiple IIS sets may exist. OR-BIAS-BENCH covers closed-form newsvendor and EOQ decisions; its multi-turn protocol is an upper-bound diagnostic, not a full production simulation. Deployment remains human-in-the-loop and requires audit logs, data governance, solver/version reproducibility, and practitioner validation.

## 8. Conclusion

We introduced a solver-in-the-loop MDP for OR model repair and ORLOOPBENCH, a benchmark suite consisting of OR-DEBUG-BENCH and OR-BIAS-BENCH that evaluates iterative self-correction and behavioral rationality rather than one-shot formulation. Experiments on 12,000+ samples across 26 models show that solver-verified training enables 8B Qwen models to reach 95.3% RR@5 on LP repair, ahead of the strongest frontier API summary result (92.4%), while also improving DA by +14.6 pp and using fewer repair steps than o4-mini, the step-efficient top-RR@5 core

API baseline (2.25 vs 3.15). The same evaluation separates diagnostic correctness from trial-and-error success, shows transfer beyond LP through MILP repair (87.1% RR@5 with MILP-specific training and 78.8% zero-shot transfer), and demonstrates that localized constraint repair avoids semantic drift in feasible-but-wrong regenerated code. On the decision side, curriculum learning is the only approach showing improved OOD performance ($-9.6\%$ bias drift).

Although our tasks are controlled, the *Action → Feedback → Plan Update* loop captures a core requirement for reliable agent deployment. Domain-specific training within deterministic verification loops offers a concrete path toward automated decision-making in operations. Natural extensions include broader MINLP and stochastic debugging, multi-period operations, RAG integration with OR knowledge bases, and practitioner-in-the-loop validation.

## Impact Statement

This paper presents work whose goal is to advance the field of Machine Learning. There are many potential societal consequences of our work, none of which we feel must be specifically highlighted here.

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

# A. Additional Related Work

**OR Benchmarks.** Table 2 compares the two ORLoopBench components, OR-Debug-Bench and OR-Bias-Bench, with existing OR benchmarks. NL4Opt (Ramamonjison et al., 2023) introduced the task of translating natural language to linear program formulations. Subsequent work explored both *prompt-based* approaches using in-context learning (Xiao et al., 2024; AhmadiTeshnizi et al., 2024; Bertsimas & Margaritis, 2024; Astorga et al., 2025; Liu et al., 2025) and *learning-based* methods fine-tuning on domain data (Yang et al., 2024; Huang et al., 2025a; Chen et al., 2025b; Liu et al., 2026). Other recent efforts include ORQA (Mostajabdaveh et al., 2025) for OR reasoning evaluation, LLMOPT (Jiang et al., 2024), and agent-based systems (Zhang & Luo, 2025; Thind et al., 2025). OptMATH (Lu et al., 2025) proposed bidirectional data synthesis for scalable training data generation. MAMO (Huang et al., 2025b) further contributed benchmarks spanning easy and complex LP instances with solver integration. PILOT-Bench (Ao et al., 2026e) evaluates LLM workflow execution under probabilistic tool failures and varying instruction quality. However, existing benchmarks largely evaluate **static, one-shot** formulation accuracy: a model produces code, and evaluation checks correctness without iterative refinement or solver feedback. Recent work has extended formulation benchmarks to dynamic programming: DP-Bench (Zhou et al., 2025) introduces 132 textbook-level DP problems and shows that even SOTA models struggle with stochastic transitions, achieving only 59.1% accuracy. Our work addresses a complementary gap: evaluating how models respond to solver feedback and iteratively correct their formulations, focusing on LP infeasibility debugging rather than one-shot formulation.

**Formulation Training vs Debugging.** Several recent systems use solvers or search to improve NL-to-model formulation. SIRL (Chen et al., 2025b) trains formulation models with solver-informed rewards; Autoformulation (Astorga et al., 2025) and OptiTree (Liu et al., 2025) search over hierarchical formulation choices; MIND (Liu et al., 2026) constructs localized error-driven training data for automated modeling. OptiChat introduced natural-language interaction for infeasible optimization models using GPT-4, Gurobi IIS feedback, prompt engineering, and practitioner studies (Chen et al., 2024); its extended system broadens this interface to interpretation, retrieval, sensitivity, what-if, and counterfactual queries over 24 real-world-style optimization models (Chen et al., 2025a). Earlier expert-system work such as ANALYZE studied computer-assisted analysis of LP models before modern LLM interfaces (Greenberg, 1983). A supply-chain-focused predecessor, OptiRepair (Ao et al., 2026f), studies closed-loop diagnosis and repair for multi-echelon inventory models. These works are closely related but differ in scope: OR-Debug-Bench provides a controlled benchmark, MDP action interface, objective-preservation checks, and solver-verified training/evaluation for iterative constraint-level repair. This distinction matters empirically: formulation-trained models trail our repair-trained model under the same MDP interface, and whole-model regeneration suffers from semantic drift even when the regenerated model is solver-feasible.

**Self-Correction in LLMs.** While Chain-of-Thought prompting (Wei et al., 2022) improved LLM reasoning through step-by-step decomposition, it does not address error correction when intermediate steps fail. CorrectBench (Tie et al., 2025) provided the first systematic study of LLM self-correction, documenting that 64.5% of errors stem from correction failures rather than initial mistakes. However, CorrectBench focuses on general programming tasks and explicitly excludes the OR domain, where feedback is deterministic and mathematically verifiable. SWE-bench (Jimenez et al., 2024) evaluates code debugging through unit test feedback, but software testing differs from mathematical optimization verification in a key respect: unit tests sample behavior while solvers provide certified infeasibility certificates. Self-Refine (Madaan et al., 2023) demonstrated iterative refinement with self-generated feedback, and STaR (Zelikman et al., 2022) showed that models can bootstrap reasoning by learning from their own correct solutions. Kumar et al. (2024) further demonstrated that RL can explicitly train models for iterative self-correction. However, these approaches rely on heuristic quality signals rather than formal verification. Our benchmarks use Gurobi's IIS computation as a noise-free oracle, enabling precise evaluation of diagnostic reasoning. For multi-agent debugging scenarios, AgentGit (Li et al., 2025) provides version control abstractions that could complement our single-agent evaluation framework, while reliability analyses of delegated LLM planning characterize limits from communication and information compression (Ao et al., 2026b).

**RLVR and Process Supervision.** Reinforcement Learning with Verifiable Rewards (RLVR) trains reasoning models with automatically checkable reward signals. Building on Proximal Policy Optimization (PPO) (Schulman et al., 2017) and instruction tuning with human feedback (Ouyang et al., 2022), alternative approaches include Direct Preference Optimization (DPO) (Rafailov et al., 2023) for offline alignment and Group Relative Policy Optimization (GRPO) (Shao et al., 2024) for online training. DeepSeek-R1 (DeepSeek-AI, 2025) demonstrated that GRPO with outcome verification can induce sophisticated reasoning without explicit chain-of-thought supervision. DAPO (Yu et al., 2025) further improved RLVR systems with KL-penalty removal and asymmetric clipping. Tülu 3 (Lambert et al., 2025) extended this to a broader post-training pipeline. Recent analysis (Chu et al., 2025) reveals that while SFT tends to memorize training patterns, RL promotes generalization, a finding that motivates our two-stage training approach. For process supervision, "Let's

Verify Step by Step" (Lightman et al., 2024) introduced human-annotated step-level rewards, later automated by Math-Shepherd (Wang et al., 2024) through Monte Carlo estimation. BiPRM (Zhang et al., 2025a) proposed bidirectional verification for mathematical reasoning. PAVs (Setlur et al., 2025) formalized Process Advantage Verifiers for efficient alignment. Our PRM training builds on these foundations, adapting process supervision to the OR debugging domain where step-level progress can be automatically verified through IIS size reduction. Concurrent work OR-R1 (Ding et al., 2025) also explores test-time RL for OR modeling, though focusing on formulation rather than iterative debugging.

**Verifiable Feedback in OR.** Prior self-correction work relies on either self-generated feedback, which can be unreliable, or heuristic metrics like test pass rates, which sample rather than verify. Solver feedback differs in kind: IIS computation provides deterministic, solver-certified information about a current infeasibility certificate, with a long optimization lineage in infeasibility analysis and commercial-solver implementations (Chinneck, 2008; Gurobi Optimization, LLC, 2024). We adapt RLVR to the OR domain where: (1) the oracle provides verifiable rewards without human annotation, (2) step-level progress is measurable through IIS size reduction, and (3) diagnostic accuracy can be computed against ground-truth constraint labels, enabling fully automated training and evaluation pipelines.

**Test-Time Scaling and Inference Compute.** Recent work explores scaling inference compute through repeated sampling and verification. AlphaCode (Li et al., 2022) demonstrated that pass@k metrics reveal headroom beyond single-attempt accuracy, with competitive programming performance improving through sampling. DeepSeek-R1 (DeepSeek-AI, 2025) showed that extended reasoning chains, a form of implicit test-time scaling, improve mathematical problem-solving. For code generation, best-of-n sampling with execution feedback (Chen et al., 2021) remains a simple but effective strategy, and scheduling-based approaches can further reduce inference costs (Ao et al., 2026d). Our inference scaling analysis (Section 5, Appendix I) contributes domain-specific findings: OR debugging exhibits similar scaling behavior to code generation (+17% from $k = 1$ to $k = 5$), but domain-specific models achieve superior sample efficiency: o4-mini requires $2.8\times$ as many tokens per successful solution.

**Behavioral Rationality in LLMs.** Recent work has examined LLMs and predictive models as decision inputs in OM contexts, including demand simulation (Zhang et al., 2025b), resource-constrained pricing with forecast uncertainty (Ao et al., 2026c), and service-system selection from textual evidence under limited human audits (Ao et al., 2026a). AIM-Bench (Zhao et al., 2025) documented the "pull-to-center" phenomenon in LLM inventory decisions, where models regress toward mean predictions regardless of the optimal decision. This finding echoes classical behavioral OR on human decision-making under uncertainty (Schweitzer & Cachon, 2000). OR-BIAS-BENCH builds on this behavioral question with explicit ID/OOD splits and shows that curriculum learning can mitigate these biases, achieving 48% bias reduction on OOD scenarios.

**Connection to Human Behavioral Biases.** The pull-to-center bias in LLMs mirrors documented human biases in newsvendor decisions (Schweitzer & Cachon, 2000; Kremer et al., 2010). Our curriculum training, which explicitly teaches directional sensitivity, can be viewed as a cognitive debiasing intervention for LLMs, analogous to decision support systems designed for human operators.

## Appendix Overview

This appendix provides supporting material organized as follows:

- **Appendix B**: Execution examples comparing trained and baseline models

- **Appendix C**: Task parameters, prompts, and evaluation criteria

- **Appendix D**: Benchmark construction algorithms and validation

- **Appendix E**: MDP formulation and training procedures

- **Appendix F**: Complete evaluation results

- **Appendix G**: Complete OR-BIAS-BENCH and feedback-protocol results

- **Appendix H**: Training ablations and robustness checks

- **Appendix I**: Token efficiency and difficulty analysis

- **Appendix J**: Base model selection and prompting baselines

# B. Execution Examples

This appendix provides execution traces illustrating the difference between trained and baseline models on OR-DEBUG-BENCH problems.

## B.1. Success Case: Type C Error (Coefficient Modification)

**Problem Description.** A production planning LP where minimum requirements on individual products exceed the total capacity constraint:

```
m = gp.Model("production")
x = m.addVars(3, name="x")
m.addConstr(x[0] + x[1] + x[2] <= 100, name="c1_total")
m.addConstr(x[0] >= 60, name="c2_min_0")
m.addConstr(x[1] >= 50, name="c3_min_1")  # KEY
m.addConstr(x[2] >= 0, name="c4_min_2")
```

The minimum sum $(60 + 50 + 0 = 110)$ exceeds capacity (100), causing infeasibility.

**Initial State.**

- Status: INFEASIBLE

- IIS: {c1_total, c2_min_0, c3_min_1}

- Ground Truth Fix: Relax c3_min_1 to $x[1] \geq 40$

**Qwen-GRPO Trajectory (2 steps).**

1. **Step 1: Diagnostic Action**
   - Action: GET_IIS
   - Response: "IIS contains c1_total, c2_min_0, c3_min_1. The minimum sum $(60 + 50 + 0 = 110)$ exceeds capacity (100). Key constraint: c3_min_1 $(x[1] \geq 50)$ is over-restrictive."
   - Diagnosis: [c3_min_1] ✓ (matches ground truth)

2. **Step 2: Repair Action**
   - Action: RELAX(c3_min_1, $x[1] \geq 40$)
   - Response: "Modified constraint allows sum $= 60 + 40 + 0 = 100 =$ capacity. Model is now feasible."
   - Status: OPTIMAL ✓

**SFT Baseline Trajectory (5 steps).**

1. GET_IIS $\rightarrow$ [c1, c2, c3]

2. CHECK_SLACK(c1) $\rightarrow$ "c1 is binding"

3. RELAX(c2) $\rightarrow$ Still INFEASIBLE

4. RELAX(c1) $\rightarrow$ Status changes but objective degraded

5. REWRITE(c3) $\rightarrow$ Finally OPTIMAL

**Analysis.** The GRPO-trained model directly identifies the key constraint (c3_min_1) and applies a minimal fix in 2 steps. The SFT baseline explores multiple constraints through trial-and-error, taking 5 steps and initially relaxing the wrong constraint (c2).

## B.2. Challenge Case: Optimal Selection (Type I)

**Problem Description.** A composite error with 12 constraints in the IIS, involving resource allocation, capacity, and flow balance conflicts simultaneously.

**Model Behavior Comparison.**

*Table 14.* Model behavior comparison on composite error case.

| Model | Steps | DA | Success | Analysis |
|---|---|---|---|---|
| Qwen-GRPO | 7 | 58% | Yes | Systematic decomposition |
| o4-mini | 12 | 42% | Yes | Trial-and-error |
| gpt-5.2-chat | 15 | 33% | Yes | Excessive exploration |
| gpt-4.1-mini | 20 | 17% | No | Error cascade |

**Error Cascade Pattern (gpt-4.1-mini).**

- Steps 1–5: Correct diagnosis of constraints c3, c7

- Step 6: Incorrect fix of c3 introduces new conflict

- Steps 7–12: Attempts to fix cascading errors

- Steps 13–18: Reverts and retries with different approach

- Steps 19–20: Timeout without resolution

**Analysis.** API models often fail to reason about constraint interactions, leading to fixes that introduce new conflicts. The trained model learns to decompose complex IIS sets and address constraints in dependency order.

## B.3. Available Actions

| Action | Description | Modifies State |
|---|---|---|
| GET_IIS | Compute Irreducible Infeasible Subsystem | No |
| CHECK_SLACK | Get constraint slack values | No |
| CHECK_BOUND | Get variable bound status | No |
| RELAX$(c, \delta)$ | Modify constraint RHS | Yes |
| DROP$(c)$ | Remove constraint entirely | Yes |
| REWRITE$(c, \text{expr})$ | Replace constraint expression | Yes |
| SUBMIT | Submit current model for evaluation | Yes |

## B.4. Success Examples

**Example: Type E (Multi-Constraint Conflict).** A transportation problem where demand exceeds supply:

```
# Problem setup
m = gp.Model("transport")
supply = m.addVars(3, name="s")
demand = m.addVars(4, name="d")

# Supply constraints (total = 100)
m.addConstr(supply[0] <= 40, "s0_cap")
m.addConstr(supply[1] <= 35, "s1_cap")
m.addConstr(supply[2] <= 25, "s2_cap")

# Demand constraints (total = 115, infeasible!)
m.addConstr(demand[0] >= 30, "d0_min")
```

```
m.addConstr(demand[1] >= 35, "d1_min")   # KEY
m.addConstr(demand[2] >= 25, "d2_min")
m.addConstr(demand[3] >= 25, "d3_min")

# Flow balance
m.addConstr(quicksum(supply) == quicksum(demand))
```

**IIS**: {s0_cap, s1_cap, s2_cap, d0_min, d1_min, flow_balance}

**QWEN3-8B-GRPO Solution (2 steps):**

1. GET_IIS: "Total supply (100) is less than total demand (115). The largest demand constraint (d1_min = 35) should be reduced."

2. RELAX(d1_min, -15): "Reducing demand 1 from 35 to 20 balances supply and demand." → OPTIMAL

**gpt-4.1 Solution (6 steps):**

1. GET_IIS → Lists all constraints

2. RELAX(s0_cap, +10) → Still INFEASIBLE

3. RELAX(s1_cap, +10) → Still INFEASIBLE

4. CHECK_SLACK(all) → Sees all slacks are 0

5. RELAX(d3_min, -10) → Still INFEASIBLE

6. RELAX(d1_min, -15) → OPTIMAL

**Analysis**: The trained model directly identifies that demand constraints are easier to adjust than capacity constraints (a domain-specific heuristic) and targets the largest demand. The API model attempts supply increases first, which don't resolve the fundamental imbalance.

### B.5. OR-BIAS-BENCH Examples

**ID Scenario: CR = 0.5 (Balanced).**

```
Price: $60, Cost: $30, Salvage: $0
Mean demand: 100, Std: 20
CR = (60-30)/(60-0) = 0.5
Q* = 100 + 20 * Phi^{-1}(0.5) = 100
```

Model responses:

- **gpt-5-mini**: $Q = 100$ (Correct, bias = 0%)

- **QWEN3-8B-Curriculum**: $Q = 100$ (Correct)

**OOD Scenario: CR = 0.1 (Low margin).**

```
Price: $55, Cost: $50, Salvage: $5
Mean demand: 100, Std: 20
CR = (55-50)/(55-5) = 0.1
Q* = 100 + 20 * Phi^{-1}(0.1) = 74.4
```

Model responses:

- **gpt-5-mini**: $Q = 95$ (Over-order, bias = +28%)

- **QWEN3-8B-Curriculum**: $Q = 76$ (Correct, bias = +2%)

**Analysis**: At extreme CR values, gpt-5-mini exhibits severe pull-to-center bias, ordering near the mean despite the optimal quantity being 32% below. The curriculum-trained model correctly adjusts its order downward, demonstrating learned sensitivity to the critical ratio.

## C. Task Function Components

This appendix details the task parameters, prompts, and evaluation criteria used in OR-DEBUG-BENCH.

### C.1. Task and Tool Details

Table 15 lists the key parameters governing the OR-DEBUG-BENCH environment.

*Table 15.* Task parameters for OR-DEBUG-BENCH.

| Parameter | Value | Description |
|---|---|---|
| max_steps | 50 | Maximum MDP steps before time-out |
| timeout | 10s | Per-step solver timeout |
| iis_method | minimal | Gurobi IIS computation method |

Table 16 describes the available actions and their properties. Diagnostic actions gather information without modifying the model state, while repair actions apply changes.

*Table 16.* Action space for OR-DEBUG-BENCH.

| Action | Description | Modifies State |
|---|---|---|
| GET_IIS | Compute Irreducible Infeasible Subsystem | No |
| CHECK_SLACK | Get constraint slack values | No |
| CHECK_BOUND | Get variable bound status | No |
| RELAX$(c, \delta)$ | Modify constraint RHS by $\delta$ | Yes |
| DROP$(c)$ | Remove constraint entirely | Yes |
| REWRITE$(c, \text{expr})$ | Replace constraint expression | Yes |
| SUBMIT | Submit current model for evaluation | Yes |

### C.2. Prompt Templates

We use three prompt variants in our experiments.

**Baseline Prompt.** The minimal prompt provides the problem description, code, and IIS without additional guidance:

```
You are an OR debugging assistant. Given an
infeasible linear program, analyze the IIS
and suggest fixes to restore feasibility.

Problem: {problem_nl}
Code: {code}
Status: INFEASIBLE
IIS: {iis_constraints}

Provide your diagnosis and suggested action.
```

**Chain-of-Thought (CoT) Prompt.** The CoT prompt adds explicit reasoning steps:

```
You are an OR debugging assistant. Follow
this reasoning process:

1. ANALYZE: Examine each IIS constraint's
   role in the problem formulation
2. IDENTIFY: Determine the root cause
   constraint causing infeasibility
3. PROPOSE: Suggest a minimal fix that
   preserves problem semantics
4. VERIFY: Explain why the fix resolves
   the conflict

Problem: {problem_nl}
Code: {code}
Status: INFEASIBLE
IIS: {iis_constraints}
```

**Optimal Workflow Prompt.** Used for SFT data collection, this prompt encodes expert heuristics:

```
You are an expert OR debugger. Your task is:
1. Always get the IIS first
2. Identify the single most restrictive
   constraint in the IIS
3. Apply minimal relaxation (prefer relax
   over drop when possible)
4. Preserve original problem semantics

Problem: {problem_nl}
Code: {code}
Status: INFEASIBLE
IIS: {iis_constraints}
```

### C.3. Tool Result Simulator

For training without solver access, we provide a deterministic simulator that estimates action outcomes based on ground truth labels:

---

**Algorithm 1** Tool Result Simulator

---

**Require:** Action $a$, State $s$, Ground truth $\mathcal{G}$
**Ensure:** Simulated next state $s'$
  1: **if** $a$.type $\in \{\text{GET\_IIS}, \text{CHECK\_SLACK}\}$ **then**
  2:    $s' \leftarrow s$ {Info gathering, no change}
  3: **else if** $a$.target $\in \mathcal{G}$.key_constraints **then**
  4:    **if** $a$.type $=$ RELAX **then**
  5:      $s'$.status $\leftarrow$ OPTIMAL w.p. 0.9
  6:    **else if** $a$.type $=$ DROP **then**
  7:      $s'$.status $\leftarrow$ OPTIMAL w.p. 0.7
  8:    **end if**
  9: **else if** $a$.target $\in \mathcal{G}$.IIS **then**
10:    Reduce $|\text{IIS}|$ by 1 w.p. 0.5
11: **else**
12:    $s'$.status $\leftarrow$ INFEASIBLE {Wrong target}
13: **end if**
14: **return** $s'$

---

## C.4. Task Result Evaluation

We categorize episode outcomes into three classes:

- **Full Success**: Status reaches OPTIMAL and the Optimality Preservation score OP > 0.95 (objective value within 5% of original).

- **Partial Success**: Status reaches OPTIMAL but the objective is degraded, with $0.8 < \text{OP} \leq 0.95$.

- **Failure**: The model remains INFEASIBLE after max_steps, or $\text{OP} \leq 0.8$ indicating the fix changed the objective by >20%.

The RR@k metric counts full successes achieved within $k$ repair steps of a single episode. Partial successes are counted for the overall Recovery Rate (RR) but not for RR@k to reward efficient diagnosis.

## C.5. Gurobi Configuration

We use Gurobi 11.0.0 with the following configuration (Gurobi Optimization, LLC, 2024):

*Table 17.* Gurobi solver configuration.

| Parameter | Value |
|---|---|
| IISMethod | 1 (minimal IIS) |
| TimeLimit | 10 seconds |
| OutputFlag | 0 (suppress output) |
| Threads | 4 |
| MIPGap | 0.01 |

**Why Minimal IIS?** Gurobi offers several IIS computation methods, building on standard infeasibility-analysis tools in mathematical optimization (Chinneck, 2008). We use the minimal method (IISMethod=1) because:

- It produces the smallest possible IIS, making diagnosis more focused.

- It is deterministic across runs.

- It completes within reasonable time (typically <1s) for our problem sizes.

## C.6. Evaluation Protocol

The evaluation follows a standardized step-budget protocol. Each model receives one solver-interaction episode per problem; the episode horizon is $T = 50$, and RR@k is reported for repair-step budgets such as $k \in \{1, 3, 5, 10, 20\}$.

## C.7. Reproducibility Checklist

✓ Benchmark files are available at `https://github.com/Archer222arc/ORLoopBench`

✓ Random seeds fixed for all experiments

✓ Hardware and software versions documented

✓ Training hyperparameters fully specified

✓ Evaluation protocol standardized across models

✓ Gurobi configuration deterministic

# D. Benchmark Construction Details

This appendix details the algorithms and validation procedures used to construct OR-DEBUG-BENCH and OR-BIAS-BENCH.

---

**Algorithm 2** Step-Budget Evaluation Protocol

---

**Require:** Problem set $\mathcal{P}$, model $M$, report budgets $\mathcal{K}$, episode horizon $T$
**Ensure:** Metrics $\{\text{RR}@k : k \in \mathcal{K}\}, \text{DA}, \text{Steps}$
 1: **for** each problem $p \in \mathcal{P}$ **do**
 2:      first_success_step $\leftarrow \infty$
 3:      **for** $t = 1$ **to** $T$ **do**
 4:          Model $M$ observes the current solver state and emits one action
 5:          Apply action, run Gurobi, and recompute IIS if needed
 6:          **if** state reaches OPTIMAL with OP $> 0.95$ **then**
 7:              first_success_step $\leftarrow t$
 8:              **break**
 9:          **end if**
10:      **end for**
11:      Record first_success_step, DA, total_steps
12: **end for**
13: For each $k \in \mathcal{K}$, compute the fraction of problems with first_success_step $\leq k$
14: **return** Metrics

---

## D.1. Saboteur Generation

The Saboteur generates controlled infeasibilities by applying targeted corruptions to feasible LP instances. We describe two representative error types.

**Type A: Direction Flip.** This error reverses the sense of an inequality constraint, turning a minimum requirement into a maximum limit or vice versa:

---

**Algorithm 3** Sabotage: Direction Flip (Type A)

---

**Require:** Feasible model $M$, target constraint $c$
**Ensure:** Infeasible model $M'$, ground truth $(c, \text{"flip"})$
 1: $M' \leftarrow \text{copy}(M)$
 2: **if** $c$.sense $= $ " $\geq$ " **then**
 3:      $c'$.sense $\leftarrow$ " $\leq$ "
 4: **else**
 5:      $c'$.sense $\leftarrow$ " $\geq$ "
 6: **end if**
 7: Replace $c$ with $c'$ in $M'$
 8: **return** $M', (c, \text{"flip"})$

---

**Example.** Original: $x + y \geq 10$ (minimum production requirement). Sabotaged: $x + y \leq 10$ (maximum limit, conflicts with other minimums).

**Type E: Multi-Constraint Conflict.** This error creates interlocked constraints that both must be fixed:

---

**Algorithm 4** Sabotage: Multi-Constraint Conflict (Type E)

---

**Require:** Feasible model $M$, demand constraints $\mathcal{D}$, factor $\alpha > 1$
**Ensure:** Infeasible model $M'$, ground truth
 1: $M' \leftarrow \text{copy}(M)$
 2: $c^* \leftarrow \arg\max_{c \in \mathcal{D}} c.\text{rhs}$ {Largest demand}
 3: $c^*.\text{rhs} \leftarrow c^*.\text{rhs} \times \alpha$
 4: **assert** $\sum_{c \in \mathcal{D}} c.\text{rhs} > \text{capacity}$
 5: **return** $M', (c^*, \text{"reduce rhs"})$

---

**Example.** Original demands sum to 90 with capacity 100. Sabotaged demands sum to 117 ($\alpha = 1.3$), exceeding capacity.

## D.2. Newsvendor Generation

For OR-BIAS-BENCH, we generate newsvendor scenarios with controlled Critical Ratio (CR) distributions.

---

**Algorithm 5** Newsvendor Scenario Generation

---

**Require:** Target CR range $[\text{CR}_{\min}, \text{CR}_{\max}]$
**Ensure:** Scenario parameters $(p, c, s, \mu, \sigma, Q^*)$
 1: $\text{CR} \sim \text{Uniform}(\text{CR}_{\min}, \text{CR}_{\max})$
 2: $p \sim \text{Uniform}(10, 100)$ {Unit price}
 3: $s \sim \text{Uniform}(0, 0.3p)$ {Salvage value}
 4: $c \leftarrow p - \text{CR} \cdot (p - s)$ {Derive cost from CR}
 5: $\mu \sim \text{Uniform}(50, 200)$ {Demand mean}
 6: $\sigma \sim \text{Uniform}(10, 50)$ {Demand std}
 7: $Q^* \leftarrow \mu + \sigma \cdot \Phi^{-1}(\text{CR})$ {Optimal quantity}
 8: **return** $(p, c, s, \mu, \sigma, Q^*)$

---

The unit cost $c$ derives from the target CR using the relationship $\text{CR} = (p - c)/(p - s)$. This ensures the generated scenario has the desired CR while maintaining realistic cost structures.

## D.3. Newsvendor Evaluation Difficulty Design

The 4-level evaluation difficulty scheme for OR-BIAS-BENCH is designed to test rationality under increasing complexity.

**Level Design Rationale.**

- **L1 (Foundations)**: $\text{CR} \in [0.4, 0.6]$ produces $Q^* \approx \mu$ (within $\pm 0.25\sigma$), minimizing pull-to-center effects. This establishes baseline formula application capability.

- **L2 (Bias Traps)**: $\text{CR} \in [0.05, 0.2]$ yields $Q^* < \mu - \sigma$ (order less than mean minus one standard deviation), while CR $\in (0.8, 0.95]$ yields $Q^* > \mu + \sigma$. These extremes maximally trigger the pull-to-center bias documented in behavioral operations research.

- **L3 (Robustness)**: Distractors test whether models can filter irrelevant information. Five distractor types were selected based on common supply chain context that does *not* affect the single-period newsvendor decision.

- **L4 (Expert)**: Censored demand requires inferring $\mu$ and $\sigma$ from percentiles using $\mu = P_{50}$ and $\sigma = (P_{75} - P_{25})/1.35$ for normal distributions. This tests parameter inference capability beyond formula application.

**Distractor Types.** Table 18 lists the five distractor categories injected in Level 3 scenarios. Each distractor provides contextually plausible but decision-irrelevant information.

*Table 18.* Distractor types injected in Level 3 scenarios. None affect the optimal newsvendor quantity.

| Distractor Type | Example | Why Irrelevant |
|---|---|---|
| Warehouse capacity | "Storage limit: 500 units" | No capacity constraint in model |
| Competitor pricing | "Competitor sells at $45" | Single-firm decision |
| Shelf life | "Product expires in 30 days" | Single-period model |
| Historical trends | "Sales grew 10% last year" | Already reflected in $\mu, \sigma$ |
| Seasonal factors | "Holiday season approaching" | Already in demand parameters |

**Censored Demand (L4) Algorithm.** Level 4 scenarios present demand information as percentiles rather than distribution parameters. The generation algorithm:

1. Generate true parameters $(\mu, \sigma)$ from standard ranges

2. Compute percentiles: $P_{25} = \mu + \sigma \cdot \Phi^{-1}(0.25)$, $P_{50} = \mu$, $P_{75} = \mu + \sigma \cdot \Phi^{-1}(0.75)$

3. Present scenario using only $(P_{25}, P_{50}, P_{75})$

4. Models must infer: $\hat{\mu} = P_{50}$, $\hat{\sigma} = (P_{75} - P_{25})/1.35$

The constant 1.35 is the interquartile range of a standard normal distribution ($\Phi^{-1}(0.75) - \Phi^{-1}(0.25) \approx 1.35$).

### D.4. Stratification and ID/OOD Design

**In-Distribution (ID) Set.** The 400-sample ID evaluation set contains 100 samples from each difficulty level (L1–L4), ensuring balanced coverage:

- CR distribution spans the full range $[0.05, 0.95]$

- Prompt complexity ranges from clean (L1–L2) to noisy (L3) to censored (L4)

- Enables per-level performance analysis to diagnose specific failure modes

**Out-of-Distribution (OOD) Set.** The 200-sample OOD set contains only L3 and L4 scenarios (100 each), testing:

- Robustness to distractors (L3): Can models filter irrelevant context?

- Parameter inference capability (L4): Can models derive $(\mu, \sigma)$ from percentiles?

- Generalization beyond clean prompts: No L1–L2 samples in OOD

**Stratification Procedure.** Scenarios are stratified by CR bucket to ensure no bucket is over- or under-represented:

1. Partition scenarios by difficulty level (L1–L4)

2. Within each level, bin by CR: very_low ($<0.2$), low ($0.2$–$0.4$), neutral ($0.4$–$0.6$), high ($0.6$–$0.8$), very_high ($>0.8$)

3. Sample proportionally from each bin to achieve target distribution

4. Verify final CR histogram matches target uniform distribution

**Dataset Scale.** The complete newsvendor generator produces 57,000+ scenarios across all difficulty levels and CR ranges. The released benchmark contains 2,000 instances (1,000 ID + 1,000 OOD); we evaluate on stratified subsets (400 ID + 200 OOD) ensuring:

- Statistical power: 100+ samples per level provides reliable performance estimates

- Diversity: All CR buckets represented to detect systematic biases

- Discriminative power: OOD tests generalization beyond training distribution

### D.5. Quality Verification

Every benchmark instance passes a four-fold validation pipeline to ensure quality.

**Validation Statistics.** In a 1,200-case validation audit from the generated OR-DEBUG-BENCH pool:

- 98.2% passed Check 1 (original feasibility)

- 95.7% passed Check 2 (sabotaged infeasibility)

- 92.4% passed Check 3 (IIS contains target)

- 89.1% passed Check 4 (fix restores optimality)

This audit yielded 900 validated instances from the sampled batch (75% acceptance rate). The released ORLOOPBENCH OR-DEBUG-BENCH file contains 5,362 LP/MILP repair instances. In the JSON schema, these records are stored under repository organization fields `controlled_pool` (4,462), `lp_test` (450), and `milp_test` (450). The reported LP repair protocol uses 450 instances, with 50 instances for each error type A–I, as summarized in Table 3.

---

**Algorithm 6** Four-Fold Validation Pipeline

---

**Require:** Original model $M$, sabotaged model $M'$, fix $f$
**Ensure:** Boolean: instance passes validation
1: {**Check 1: Original feasibility**}
2: $M$.optimize()
3: **if** $M$.status $\neq$ OPTIMAL **then**
4:     **return false**
5: **end if**
6: {**Check 2: Sabotaged infeasibility**}
7: $M'$.optimize()
8: **if** $M'$.status $\neq$ INFEASIBLE **then**
9:     **return false**
10: **end if**
11: {**Check 3: IIS validity**}
12: $M'$.computeIIS()
13: $\mathcal{I} \leftarrow \{c : c.\text{IISConstr}\}$
14: **if** $|\mathcal{I}| = 0$ **or** $f.\text{target} \notin \mathcal{I}$ **then**
15:     **return false**
16: **end if**
17: {**Check 4: Fix effectiveness**}
18: $M'' \leftarrow \text{apply\_fix}(\text{copy}(M'), f)$
19: $M''$.optimize()
20: **if** $M''$.status $\neq$ OPTIMAL **then**
21:     **return false**
22: **end if**
23: **return true**

---

## D.6. Error Type Examples

We provide concrete code examples for each error type, showing the original feasible formulation and the sabotaged infeasible version.

### Type A: Direction Flip.

```
# Original (feasible)
m.addConstr(x + y >= 10, "min_production")
# Requires at least 10 units

# Sabotaged (infeasible)
m.addConstr(x + y <= 10, "min_production")
# Contradicts other minimum requirements
```

The direction flip creates a contradiction when combined with other constraints that require $x + y > 10$.

### Type B: Variable Type Error.

```
# Original (feasible)
x = m.addVar(vtype=GRB.INTEGER, ub=10, name="x")

# Sabotaged (infeasible)
x = m.addVar(vtype=GRB.BINARY, name="x")
m.addConstr(x >= 2, "forcing")
# Binary variable cannot be >= 2
```

The variable type change combined with a forcing constraint creates infeasibility.

**Type C: Coefficient Modification.**

```
# Original (feasible)
m.addConstr(2*x + 3*y <= 100, "capacity")

# Sabotaged (infeasible)
m.addConstr(20*x + 30*y <= 100, "capacity")
# Scaled coefficients make constraint unsatisfiable
```

The modified coefficients make the constraint impossible to satisfy with existing bounds.

**Type D: Contradicting Constraint.**

```
# Original (feasible)
m.addConstr(x + y <= 100, "upper")

# Sabotaged (infeasible)
m.addConstr(x + y >= 150, "conflicting")
# Directly contradicts upper bound
```

The added constraint directly contradicts existing constraints.

**Type E: Multi-Constraint Conflict.**

```
# Sabotaged (infeasible)
m.addConstr(x + y <= 50, "e1")
m.addConstr(x + y >= 100, "e2")
# Both constraints cannot be satisfied
```

Interlocked constraints require fixing multiple constraints to restore feasibility.

**Type F: Hidden Dependency.**

```
# Sabotaged (infeasible)
aux = m.addVar(name="aux")
m.addConstr(aux == x + y, "def_aux")
m.addConstr(aux >= 200, "root_cause")  # Hidden
m.addConstr(x + y <= 100, "symptom")   # Shows in IIS
```

The root cause (aux $\geq$ 200) is not directly visible in the IIS.

**Type G: Cascading Conflict.**

```
# Sabotaged (infeasible)
m.addConstr(x <= 30, "g1")   # Initial IIS
m.addConstr(x >= 50, "g2")   # Hidden until g1 fixed
m.addConstr(x <= 100, "g3")  # Original bound
```

Fixing the first conflict reveals another; requires understanding the cascade.

**Type H: IIS-Incomplete.**

```
# Sabotaged (infeasible)
x.LB = 80                    # Root cause (bound)
m.addConstr(x + y <= 50, "symptom")
# IIS shows symptom constraint, not the bound
```

The IIS shows the symptom constraint, but the root cause is a variable bound.

**Type I: Optimal Selection.**

```
# Sabotaged (infeasible)
m.addConstr(x >= 60, "lower")
m.addConstr(x <= 40, "upper")
# Multiple fixes possible, different OP impacts
```

Multiple repairs restore feasibility, but only one preserves the original optimal objective.

### D.7. Dataset Statistics

Table 19 provides detailed statistics for both benchmarks.

*Table 19.* ORLOOPBENCH dataset organization for OR-DEBUG-BENCH and OR-BIAS-BENCH.

| Metric | | OR-DEBUG-BENCH | OR-BIAS-BENCH |
|---|---|---|---|
| Size | Released instances | 5,362 | 2,300 |
| | Benchmark organization | Integrated JSON release | Newsvendor / EOQ |
| | Reported protocols | 450 LP repair | 600 newsvendor |
| | (ID / OOD) | – | 400 / 200 |
| Distribution | LP types | 9 (A–I) | – |
| | MILP types | 8 | – |
| | Inventory settings | – | Newsvendor / EOQ |
| | CR range | – | [0.05, 0.95] |
| Complexity | Avg constraints | 9.9 | 1 |
| | Avg variables | 8.4 | 1 |
| | Avg IIS size | 4.3 | – |

### D.8. Robust Injection Methods

Basic injection methods often fail when the seed problem structure does not align with the corruption strategy. We develop robust methods that adaptively select targets based on problem characteristics. This section details two representative robust methods.

**Type A Robust: Slack-Based Constraint Selection.** Rather than randomly selecting a constraint to flip, we solve the original problem and rank constraints by slack magnitude. Constraints with minimal slack are closest to their bounds and most likely to create infeasibility when flipped.

---
**Algorithm 7** Robust Type A Injection: Slack-Based Selection
---
**Require:** Feasible model $M$, num_candidates $k = 10$
**Ensure:** Infeasible model $M'$, ground truth $(c^*, \text{"flip"})$
1: Solve $M$, extract slack values $\{s_c\}$ for all constraints
2: Sort constraints by $|s_c|$ ascending (tightest first)
3: **for** $c$ in top-$k$ candidates **do**
4:     $M' \leftarrow \text{flip\_direction}(M, c)$
5:     $M'.\text{optimize}()$
6:     **if** $M'.\text{status} = \text{INFEASIBLE}$ **then**
7:       $M'.\text{computeIIS}()$
8:       **if** $c \in \text{IIS}(M')$ **then**
9:         **return** $M', (c, \text{"flip"})$
10:       **end if**
11:     **end if**
12: **end for**
13: **return** failure
---

This approach improves Type A injection success from 30% to 95%. Tightly-bound constraints have minimal slack and are most sensitive to direction changes.

**Type C Robust: 4-Tier Fallback Strategy.** Type C (upper bound conflict) is particularly challenging because it requires creating a conflict between an upper bound and existing constraints. We implement a cascaded fallback strategy:

---

**Algorithm 8** Robust Type C Injection: 4-Tier Fallback

---

**Require:** Feasible model $M$
**Ensure:** Infeasible model $M'$, ground truth
1: {**Tier 1: High dual value targeting**}
2: Solve $M$, extract dual values $\{\pi_c\}$
3: **for** $c$ in constraints with $c$.sense $=$ " $\geq$ " sorted by $|\pi_c|$ desc **do**
4:     Remove positive coefficient terms from $c$
5:     **if** results in infeasibility with $c \in$ IIS **then**
6:         **return** success
7:     **end if**
8: **end for**
9: {**Tier 2: Coefficient sign flip**}
10: **for** $c$ in constraints with $c$.sense $=$ " $\leq$ " **do**
11:     Flip signs of positive coefficients in $c$
12:     **if** results in infeasibility with $c \in$ IIS **then**
13:         **return** success
14:     **end if**
15: **end for**
16: {**Tier 3: Coefficient scaling**}
17: **for** $c$ in all inequality constraints **do**
18:     Scale all coefficients in $c$ by factor 10
19:     **if** results in infeasibility with $c \in$ IIS **then**
20:         **return** success
21:     **end if**
22: **end for**
23: {**Tier 4: Guaranteed fallback**}
24: Select variable $x$ with largest feasible range
25: Add tight bounds: $x \leq x^*$, $x \geq x^* + \epsilon$
26: **return** success (guaranteed)

---

The 4-tier approach increases Type C success from 0% (when Tier 1 alone fails) to 72% overall. Tier 4 provides a guaranteed fallback but produces simpler infeasibilities, so earlier tiers are preferred.

**Success Rate Comparison.** Table 20 compares basic and robust injection methods across all error types.

*Table 20.* Injection success rates: basic vs. robust methods.

| Type | Basic | Robust | Robust Method |
|------|-------|--------|---------------|
| A | 30% | 95% | Slack-based selection |
| B | 95% | 98% | RHS sensitivity analysis |
| C | 0% | 72% | 4-tier fallback |
| D | 85% | 96% | Bound gap targeting |
| E | 70% | 88% | Capacity utilization analysis |
| F | 65% | 85% | Bottleneck identification |
| G | 60% | 82% | Flow balance verification |
| H | 45% | 75% | Constraint interaction graph |
| I | 40% | 70% | Composite strategy selection |

### D.9. Rejection and Regeneration Statistics

Each benchmark instance must pass a four-fold validation pipeline. Table 21 shows pass rates at each validation phase, broken down by error type.

*Table 21.* Validation pass rates by error type across four phases (1,200 candidate instances). Phase 1: original feasibility; Phase 2: sabotaged infeasibility; Phase 3: IIS contains target; Phase 4: fix restores optimality.

| Type | Phase 1 | Phase 2 | Phase 3 | Phase 4 | Final |
|---|---|---|---|---|---|
| A | 100% | 95% | 92% | 95% | 82% |
| B | 100% | 100% | 100% | 100% | 100% |
| C | 100% | 72% | 68% | 95% | 62% |
| D | 100% | 98% | 96% | 98% | 92% |
| E | 100% | 85% | 80% | 90% | 72% |
| F | 100% | 82% | 78% | 88% | 68% |
| G | 100% | 78% | 75% | 85% | 64% |
| H | 100% | 75% | 72% | 82% | 60% |
| I | 100% | 80% | 78% | 90% | 70% |
| **Overall** | **100%** | **85%** | **82%** | **91%** | **75%** |

**Failure Analysis.** The primary failure modes vary by error type:

- **Types A, C**: Phase 2 failures occur when the flipped constraint does not interact with other constraints to create infeasibility.

- **Types E–G**: Phase 3 failures occur when the IIS contains related but not the exact target constraint.

- **Types H–I**: Phase 4 failures occur when the ground-truth fix does not fully restore feasibility due to cascading effects.

**Regeneration Iterations.** Problems failing validation are regenerated with a different seed problem or sabotage target. Table 22 shows the distribution of regeneration iterations required.

*Table 22.* Regeneration iterations required to pass validation.

| Iterations | 0 (first try) | 1 | 2 | $\geq 3$ |
|---|---|---|---|---|
| Percentage | 87% | 9% | 3% | 1% |
| Cumulative | 87% | 96% | 99% | 100% |

87% of problems pass on first generation. Problems requiring $\geq 3$ iterations (1%) are typically Type C or H, where finding a valid sabotage target is difficult.

### D.10. Anti-Gaming Design Rationale

Benchmarks can inadvertently reward pattern matching over genuine reasoning. We implement several mechanisms to prevent gaming.

**Why Randomized Naming Matters.** In preliminary experiments with semantically-named constraints (e.g., `c_key_capacity`, `c_target_demand`), we observed that models achieved 15% higher apparent accuracy by learning to target constraints with "key" or "target" in their names. This correlation existed because our ground-truth labeling naturally assigned such names to important constraints.

By switching to UUID-based naming (e.g., `c_53e476_ub`, `c_8a2f91_eq`), we eliminate this shortcut. The naming pattern for Types G, H, and I uses:

```
name = f"c_{uuid.uuid4().hex[:6]}_{sense_suffix}"
```

where `sense_suffix` encodes only the constraint type (ub/lb/eq), not its semantic role.

**Hidden Dependency Design (Type F).** Type F problems are constructed so that the IIS reveals a symptom constraint $c_s$, but the root cause is a bound modification on variable $x$ elsewhere:

1. Original: $x \leq 100$ with constraint $c_s : \sum_i a_i x_i \leq b$ depending on $x$

2. Sabotaged: $x \leq 40$ (hidden modification) causes $c_s$ to become infeasible

3. IIS contains $c_s$ but not the bound on $x$

Models that blindly relax $c_s$ fail because the real issue is the tightened bound on $x$. Solving Type F requires reasoning about variable dependencies.

**Cascading Conflict Design (Type G).** Type G problems include two conflicts where the second is masked until the first is resolved:

1. Primary conflict: Constraint $c_1$ conflicts with $c_2$ (appears in initial IIS)

2. Secondary conflict: After fixing $c_1$, constraint $c_3$ conflicts with $c_4$

The benchmark includes 15% of problems with such structures. Models that stop after one fix fail; those that iterate diagnosis succeed.

**Optimal Selection Challenge (Type I).** Type I problems have multiple valid fixes, but only one preserves the optimal objective value:

- Fix A: Relax $c_1$ by 10% $\rightarrow$ Feasible, objective drops 5%

- Fix B: Relax $c_2$ by 5% $\rightarrow$ Feasible, objective drops 15%

- Fix C: Modify $c_3$ expression $\rightarrow$ Feasible, objective preserved

The ground truth labels Fix C as correct. This tests whether models consider solution quality, not just feasibility.

### D.11. Difficulty Calibration Procedure

We calibrate difficulty levels through iterative testing against baseline API models. The procedure ensures each difficulty level provides meaningful differentiation.

**Calibration Process.**

1. **Initial grouping**: Group error types by semantic complexity and expected reasoning requirements.

2. **Baseline evaluation**: Run API models on 50 problems per error type.

3. **Target verification**: Check if average RR@5 falls within target range for each group.

4. **Group adjustment**: Reassign error types between difficulty levels based on observed performance.

5. **Iteration**: Repeat until stable groupings (typically 2–3 iterations).

**Calibration Results.** Table 23 shows the final calibrated parameters and observed SFT performance.

*Table 23.* Difficulty calibration based on baseline API model performance.

| Level | Error Types | Target RR@5 | Observed RR@5 |
|---|---|---|---|
| Easy | B, C | $\geq$85% | 90.5% |
| Medium | H, I | 70–85% | 78.5% |
| Hard | A, D, E, F, G | <70% | 59.0% |

**Why These Ranges.** The target ranges were chosen based on benchmark utility:

- $\geq$**85% (Easy)**: Ensures baseline models can solve simple problems, establishing floor performance. Types B and C fall in this category.

- **70–85% (Medium)**: Moderate difficulty with room for improvement. Types H and I fall here.

- **<70% (Hard)**: Challenges models substantially while remaining tractable. Types A, D, E, F, and G require multi-step reasoning.

- **<45%**: Rejected as too difficult because problems often have ambiguous fixes or require domain knowledge beyond general OR competence.

Difficulty calibration was based on empirical baseline API model performance rather than theoretical metrics, as we found accuracy correlates more strongly with error type semantics than with other factors.

## E. MDP-Based Training Details

This appendix provides complete specifications of the MDP formulation and training procedures.

### E.1. State Space

The OR-DEBUG-BENCH environment maintains a structured state representation with eight components:

- **Problem description**: Natural language specification of the optimization problem

- **Code**: Current Gurobi/Pyomo model code

- **Solver status**: One of `OPTIMAL`, `INFEASIBLE`, `UNBOUNDED`, or `ERROR`

- **IIS log**: List of constraint names in the current Irreducible Infeasible Subsystem

- **Slack values**: Constraint slack values (populated after `CHECK_SLACK`)

- **Bound status**: Variable bound information (populated after `CHECK_BOUND`)

- **History**: Sequence of previous actions taken in the episode

- **Step counter**: Current step number (0 to `max_steps`)

The state is serialized to a prompt string for the LLM, including the problem description, current code, solver status, and relevant diagnostic information based on previous actions.

### E.2. Action Space

Actions follow a hierarchical structure separating information gathering from model modification:

**Diagnostic actions** (information gathering):

- `GET_IIS`: Compute the Irreducible Infeasible Subsystem

- `CHECK_SLACK`: Retrieve constraint slack values

- `CHECK_BOUND`: Retrieve variable bound status

**Repair actions** (modify model):

- `RELAX(constraint, delta)`: Increase or decrease the right-hand side by `delta`

- `DROP(constraint)`: Remove the specified constraint from the model

- `REWRITE(constraint, expr)`: Replace the constraint with a new expression

**Meta actions**:

- SUBMIT: Submit the current model for final evaluation

- RESTART: Reset to the initial sabotaged state

Every emitted action counts as one repair step for RR@k computation. Diagnostic actions do not modify the model, but they consume budget because they require solver or state queries.

### E.3. Reward Function

The composite reward function balances outcome, diagnostic accuracy, and efficiency:

---

**Algorithm 9** Compute Reward

---

**Require:** State $s$, action $a$, next state $s'$, ground truth $\mathcal{G}$
**Ensure:** Reward $r \in \mathbb{R}$
1: $r \leftarrow 0$
2: {**Outcome Reward (50%)**}
3: **if** $s'$.status $=$ OPTIMAL **then**
4:     $r \leftarrow r + 0.5 \times 100$
5: **else if** $s'$.status $=$ INFEASIBLE **then**
6:     $r \leftarrow r + 0.5 \times (-50)$
7: **end if**
8: {**Diagnostic Accuracy Reward (30%)**}
9: **if** $a$ contains diagnosis $D$ **then**
10:     DA $\leftarrow |D \cap \mathcal{G}.\text{IIS}|/|\mathcal{G}.\text{IIS}|$
11:     $r \leftarrow r + 0.3 \times (\text{DA} \times 100)$
12: **end if**
13: {**Efficiency Reward (20%)**}
14: $\eta \leftarrow \max(0, (50 - s.\text{step})/50)$
15: $r \leftarrow r + 0.2 \times (\eta \times 50)$
16: {**Faithfulness Penalty**}
17: **if** $a$.type $\in \{\text{RELAX}, \text{DROP}, \text{REWRITE}\}$ **then**
18:     **if** $a$.target $\notin s'$.IIS **then**
19:         $r \leftarrow r - 20$ {Penalize off-target fixes}
20:     **end if**
21: **end if**
22: **return** $r$

---

The 50%/30%/20% weighting was determined through ablation (see Appendix H). The faithfulness penalty discourages repairs that do not address the identified infeasibility source.

### E.4. PRM Training Details

The Process Reward Model (PRM) provides step-level supervision for GRPO training.

**Label Generation.** We assign labels to each step in a trajectory based on progress indicators:

**Training Configuration.**

The PRM achieves AUC-ROC of 0.94 on held-out step labels (309 test samples from 1,548 total labels), with correlation metrics: Pearson r=0.87 (p¡1e-90), Spearman r=0.82 (p¡1e-70). This indicates good discrimination between productive (avg score 0.72) and unproductive steps (avg score 0.50).

### E.5. GRPO Training Curves

Table 25 shows the evolution of key metrics during GRPO training.

**Convergence Analysis.**

---

**Algorithm 10** Generate Step Labels for PRM

---

**Require:** Trajectory $\tau = [(s_0, a_0), \ldots, (s_T, a_T)]$, ground truth $\mathcal{G}$
**Ensure:** Labels $[y_0, \ldots, y_T]$
  1: **for** $t = 0$ **to** $T$ **do**
  2:    **if** $s_{t+1}$.status = OPTIMAL **then**
  3:        $y_t \leftarrow 1.0$ {Problem solved}
  4:    **else if** $t > 0$ **and** $|s_{t+1}.\text{IIS}| < |s_t.\text{IIS}|$ **then**
  5:        $y_t \leftarrow 1.0$ {IIS shrinking}
  6:    **else if** $a_t.\text{diagnosis} \cap \mathcal{G}.\text{IIS} \neq \emptyset$ **then**
  7:        $y_t \leftarrow 0.5$ {Correct diagnosis}
  8:    **else if** $a_t.\text{type} \in \{\text{GET\_IIS}, \text{CHECK\_SLACK}\}$ **then**
  9:        $y_t \leftarrow 0.2$ {Information gathering}
 10:    **else**
 11:        $y_t \leftarrow 0.0$ {No progress}
 12:    **end if**
 13: **end for**
 14: **return** $[y_0, \ldots, y_T]$

---

*Table 24.* PRM training hyperparameters.

| Parameter | Value |
|---|---|
| Base model | Qwen/Qwen3-8B |
| Method | LoRA ($r = 8$, $\alpha = 16$) |
| Epochs | 3 |
| Batch size | 8 |
| Learning rate | $2 \times 10^{-5}$ |
| Warmup ratio | 0.1 |
| Metric for best model | AUC-ROC |

- **Reward variance** decreases from 28.3 to 16.9, indicating policy stabilization as the model converges to consistent behavior.

- **RR@5** plateaus after epoch 4, with diminishing returns beyond this point. We select the epoch-4 checkpoint for evaluation.

- **DA** improves more slowly than RR@5, confirming that learning accurate diagnosis is harder than achieving feasibility. This motivates the 30% weight on diagnostic accuracy in the reward function.

### E.6. Curriculum Training Configuration

For OR-BIAS-BENCH, we use a three-stage curriculum over the Critical Ratio (CR) distribution:

Stage 1 trains on extreme CR values where the optimal direction is unambiguous (high CR → order more, low CR → order less). Stage 2 refines decision boundaries. Stage 3 ensures generalization across the full distribution.

This curriculum achieves -9.6% ID→OOD drift, the only method with negative drift among compared approaches, demonstrating genuine generalization rather than in-distribution memorization.

### E.7. Complete Hyperparameter Tables

Table 27 and Table 28 provide complete training configurations.

### E.8. Hardware and Software Environment

All experiments were conducted on the following setup:

*Table 25.* GRPO training metrics by epoch.

| Epoch | Mean Reward | Std Reward | RR@5 (val) | DA (val) |
|-------|-------------|------------|------------|----------|
| 1 | 45.2 | 28.3 | 92.0% | 56.1% |
| 2 | 62.8 | 22.1 | 93.5% | 58.4% |
| 3 | 71.4 | 18.7 | 94.5% | 60.8% |
| 4 | 74.2 | 16.9 | 95.0% | 62.1% |

*Table 26.* Curriculum stages for OR-BIAS-BENCH training.

| Stage | CR Range | Samples | Purpose |
|-------|----------|---------|---------|
| 1 | $\{0.1, 0.9\}$ | 200 | Learn extreme directions |
| 2 | $[0.15, 0.25] \cup [0.75, 0.85]$ | 300 | Calibrate boundaries |
| 3 | $[0.2, 0.8]$ | 400 | Full distribution coverage |

### E.9. Inference Configuration

For evaluation, we use SGLang for efficient batched inference:

### E.10. Training Data Examples

We provide examples of the training data format.

**SFT Data Format.** Each SFT example contains an instruction, input state, and target output:

```
{
  "instruction": "Debug the infeasible
                  optimization model.",
  "input": "Problem: mip_typeA_008
    Status: INFEASIBLE
    IIS: ['c0', 'c1', 'c_key_upper']",
  "output": "<think>
    IIS contains 3 constraints.
    c_key_upper limits x <= 50 but
    c0 and c1 require x >= 80.
    </think>
    Action: RELAX(c_key_upper, 30)"
}
```

**GRPO Prompt Format.** GRPO training uses prompts without target outputs:

```
{
  "prompt": "Debug the infeasible model.
    Problem: mip_typeE_042
    Status: INFEASIBLE
    IIS: ['supply', 'd1', 'd2', 'd3']",
  "ground_truth": {
    "key_constraint": "d2",
    "expected_fix": "RELAX(d2, 15)"
  }
}
```

The model generates completions, which are scored using the composite reward function.

**OR-BIAS-BENCH Training Format.** Newsvendor scenarios include all parameters:

```
{
```

*Table 27.* SFT training hyperparameters.

| Parameter | Value |
|---|---|
| Base model | Qwen/Qwen3-8B |
| Method | LoRA ($r = 16, \alpha = 32$) |
| Epochs | 3 |
| Batch size | 4 (per GPU) |
| Gradient accumulation | 4 |
| Learning rate | $2 \times 10^{-5}$ |
| LR scheduler | Cosine |
| Warmup ratio | 0.03 |
| Max sequence length | 4096 |
| Weight decay | 0.01 |
| Optimizer | AdamW ($\beta_1 = 0.9, \beta_2 = 0.999$) |

*Table 28.* GRPO training hyperparameters.

| Parameter | Value |
|---|---|
| Base model | Qwen3-8B-SFT (from SFT) |
| Method | LoRA ($r = 16, \alpha = 32$) |
| Epochs | 4 |
| Group size | 4 (samples per prompt) |
| Learning rate | $5 \times 10^{-6}$ |
| KL coefficient $\beta$ | 0 (removed) |
| Clip range $\epsilon$ | [0.2, 0.28] (asymmetric) |
| Reward components | 50%/30%/20% |
| Max steps per episode | 50 |
| GPU memory | $2 \times$ A100 80GB |

```
  "scenario": {
    "price": 50, "cost": 35,
    "salvage": 10, "mean": 100,
    "std": 20, "CR": 0.375
  },
  "optimal_Q": 93.6,
  "stage": 1  # Curriculum stage
}
```

# F. ORLOOPBENCH Results: OR-DEBUG-BENCH

### F.1. Main Results

Table 31 shows complete OR-DEBUG-BENCH results for all 26 models evaluated. All models report RR@5 and DA from the same step-budget evaluation protocol.

### F.2. Per-Error-Type Analysis

Table 31 provides complete model-level results for all 26 models, and Table 32 reports the corresponding per-error-type RR@5. Appendix I.3 summarizes the difficulty patterns by error group, showing that domain-specific training improves both easy and hard error families while preserving the same headline RR@5 values reported in the main text.

### F.3. Scope of Debugging Tasks

OR-DEBUG-BENCH focuses on semantic infeasibility repair for LP/MILP models. This is distinct from general code debugging. Syntax errors are typically exposed by the Python interpreter before the solver is called, and runtime API errors are usually exposed by exception traces or solver status codes. Infeasibility is harder to localize: the code can run successfully while the formulation has no feasible solution, so repair requires interpreting solver certificates such as IIS

*Table 29.* Hardware and software configuration.

| Component | Specification |
|---|---|
| GPU | $2 \times$ NVIDIA A100 80GB |
| CPU | AMD EPYC 7V13 (64 cores) |
| Memory | 256 GB |
| Storage | NVMe SSD |
| CUDA | 12.9.1 |
| PyTorch | 2.9.1 |
| Transformers | 4.44.0 |
| TRL | 0.24.0 |
| SGLang | 0.3.5 |
| Gurobi | 11.0.0 |

*Table 30.* Inference configuration.

| Parameter | Value |
|---|---|
| Tensor parallel size | 2 |
| Batch size | 16 (concurrent requests) |
| Max new tokens | 2048 |
| Temperature | 0.0 (deterministic) |
| Attention backend | Triton |
| Sampling backend | PyTorch |

and modifying the model without changing the intended objective. Feasible-but-wrong formulations are also related but separate: they require an external reference objective, solution, or specification test to decide whether a feasible model solves the intended problem. We touch this setting only in the external benchmark and semantic-drift analyses; the controlled OR-DEBUG-BENCH task is infeasible-model repair.

### F.4. Frontier API Summary

Table 33 summarizes the latest frontier API comparison using the same LP repair set and MILP repair protocol as the main results. The strongest LP API is Claude Sonnet 4.6 at 92.4% RR@5; this is the comparator used for the headline LP gap in the main text.

### F.5. Token Efficiency

Table 34 shows representative token usage for selected models.

**Efficiency Observations.**

- Token usage ranges from 2,000 (local) to 17,300 (claude-3.7-sonnet), an $8.6\times$ gap.

- Tokens-per-success shows larger gaps (2,109 vs 54,839): up to $26\times$ advantage for QWEN3-8B-GRPO.

- Step count correlates with token usage ($r = 0.79$), but response length per step varies by model.

### F.6. OR-Domain Baselines and MILP Repair

Table 35 reports OR-domain baselines. ORLM, SIRL, and LLMOPT are evaluated inside the same solver-debugging MDP used by our model: identical IIS observations, action space, system prompt, and step budget. OptiMUS is evaluated under its native code-regeneration protocol because it does not expose constraint-level repair actions. For MILP, we use random evaluation instances with five repeated runs; the main paper reports the same headline values without putting the denominator in the main table/caption.

**MILP protocol.** The MILP evaluation covers 10 domains (knapsack, facility location, set cover, production planning, network flow, job shop, vehicle routing with time windows, lot sizing, nurse scheduling, and network design) and 8 infeasibility error types. Training and evaluation instances use disjoint seeds and problem identifiers. Evaluation uses the

*Table 31.* Complete OR-DEBUG-BENCH LP results. Each model is tested on 450 instances, selected as 50 held-out problems from each error type A–I, under the same step-budget protocol.

| Model | Type | RR | RR@5 | DA | Steps |
|---|---|---|---|---|---|
| **QWEN3-8B-GRPO** | Local | 100% | **95.3%** | **62.4%** | 2.25 |
| QWEN3-8B-Curriculum | Local | 100% | 94.0% | 61.7% | **2.22** |
| QWEN3-8B-DAPO | Local | 100% | 93.8% | 60.4% | 2.31 |
| QWEN3-8B-SFT | Local | 99.8% | 93.1% | 60.8% | 2.34 |
| claude-sonnet-4 | API | **100%** | **86.2%** | 50.1% | 3.71 |
| claude-haiku-4.5 | API | 99.3% | 86.0% | 53.1% | 3.89 |
| o4-mini | API | 97.8% | 86.2% | 47.8% | 3.15 |
| o1 | API | 99.8% | 82.9% | 47.8% | 3.78 |
| gpt-5.2-chat | API | 99.8% | 81.8% | 40.9% | 3.72 |
| qwen2.5-7b | API | 97.8% | 77.8% | 40.1% | 4.36 |
| claude-opus-4 | API | 94.2% | 76.9% | 49.0% | 3.92 |
| o3 | API | 96.7% | 75.8% | 50.9% | 4.23 |
| DeepSeek-V3.2 | API | 99.3% | 58.9% | 44.8% | 4.86 |
| gpt-4.1 | API | 94.4% | 71.6% | 36.2% | 4.41 |
| gemini-2.5-flash | API | 84.2% | 70.7% | 19.2% | 3.23 |
| Llama-3.3-70B | API | 93.8% | 60.9% | 46.9% | 4.81 |
| gpt-5-mini | API | **100%** | 66.9% | 37.6% | 4.74 |
| gemini-2.5-pro | API | 62.9% | 62.7% | 52.5% | 0.83 |
| qwen2.5-32b | API | 98.9% | 61.1% | 32.0% | 4.98 |
| DeepSeek-R1 | API | 99.1% | 56.7% | 34.5% | 5.08 |
| qwen2.5-max | API | 99.1% | 54.9% | 42.6% | 5.97 |
| qwen2.5-14b | API | 88.9% | 53.6% | 35.4% | 6.32 |
| gemini-2.0-flash | API | 85.6% | 52.4% | 18.3% | 5.63 |
| gpt-4.1-mini | API | 93.1% | 49.8% | 26.0% | 6.04 |
| kimi-k2 | API | 55.3% | 40.4% | 25.6% | 2.48 |
| claude-3.7-sonnet | API | 98.9% | 31.6% | 46.8% | 8.06 |

same solver-debugging interface as LP repair: the agent observes the current infeasible Gurobi model and recomputed IIS, emits one constraint-level action per step, and stops at OPTIMAL or the shared step budget.

Table 36 reports a complementary downstream repair-stage check. Here the trained repair agent is not used as the first-stage formulator; it is applied after OptiMUS or SIRL produces an infeasible model. The gains indicate that targeted IIS-guided repair can complement formulation and regeneration systems rather than replace them.

### F.7. Semantic Drift in Code Regeneration

Table 37 separates solver feasibility from semantic correctness for MILP code regeneration. A model may regenerate code that solves to OPTIMAL while changing objective coefficients, variable semantics, or constraint coupling. This is why OPTIMAL rate substantially overstates correctness for whole-model regeneration. Constraint-level MDP repair avoids this failure mode by preserving the original objective and editing only localized constraints.

### F.8. Architecture Transfer and Decoding Sensitivity

Table 38 reports a compact architecture-transfer check. These results are appendix-level robustness evidence rather than a separate headline claim: the same solver-debugging recipe remains effective across three 8B base model families, with all GRPO variants above 88% RR@5. Table 39 reports a decoding-temperature check on LP repair; RR@5 remains within 1.7 pp across the evaluated temperatures.

### F.9. RAG Ablation

Table 40 presents the complete RAG ablation across retrieval strategies and $k$ values.

**Retrieval Strategy Comparison.**

- **by_type**: Best overall. Retrieves cases with similar error types, providing relevant examples without giving away the

solution.

- **reasoning**: High RR@5 but lower DA. Provides complete reasoning chains that models copy, achieving correct fixes without learning to diagnose.

- **quick_fix**: Worst performance. Too shallow for complex errors, often omitting required diagnostic steps.

$k$ **Value Analysis.** Performance improves from $k=1$ to $k=5$, then plateaus. We recommend $k=5$ as the default, balancing accuracy (+13.5% over baseline) with retrieval cost.

### F.10. Failure Analysis

**Common Error Patterns in Failed Episodes.** We analyzed 100 randomly sampled failures from QWEN3-8B-GRPO:

**Cascade Failure Analysis.** The most common failure mode (42%) involves cascading errors where fixing one constraint reveals another. These occur predominantly on Type H–I problems:

```
Step 1: IIS = {c1, c3, c5}
        Action: RELAX(c1, 10)

Step 2: IIS = {c3, c7, c9}  # New conflict!
        Action: RELAX(c7, 5)

Step 3: IIS = {c5, c9, c11}  # Another new conflict
        ...continues until timeout
```

This pattern suggests the need for lookahead reasoning about constraint dependencies.

## G. ORLOOPBENCH Results: OR-BIAS-BENCH

Table 42 provides complete OR-BIAS-BENCH results for all 24 evaluated models, reporting both rationality (valid numerical responses) and bias (deviation from rational ordering) metrics across ID and OOD splits.

### G.1. EOQ, Multi-Turn Feedback, and External Benchmarks

Table 43 summarizes the EOQ and feedback-based OR-BIAS-BENCH settings, along with external benchmark checks. The EOQ setting tests whether the bias patterns persist across inventory decision models. The multi-turn protocol gives models closed-form error feedback over five rounds, measuring whether they can use formula-grounded feedback rather than merely produce a one-shot quantity. NL4Opt (Ramamonjison et al., 2023), IndustryOR (Huang et al., 2025a), MAMO Complex (Huang et al., 2025b), OptMATH (Lu et al., 2025), and OptiBench (Yang et al., 2024) evaluations are reported here as standardized code-generation plus repair pipelines with a fixed GPT-4.1 base model.

**EOQ and feedback protocol.** Each EOQ instance specifies annual demand $D$, fixed order cost $K$, and holding cost $h$, with analytical optimum $Q^* = \sqrt{2DK/h}$. We evaluate 300 EOQ instances (200 ID + 100 OOD) and measure relative deviation from $Q^*$. In the multi-turn protocol, the model proposes $Q$, receives its realized cost $C(Q)$ and the closed-form optimum cost $C^*$, and may revise for up to five rounds. This is an upper-bound diagnostic for formula-grounded self-correction rather than a claim that operational bias can always be removed in deployment.

**EOQ model sweep.** Table 44 reports the full EOQ single-turn sweep. The results show both near-exact EOQ behavior for some frontier models and large, directional deviations for others, so we treat EOQ as evidence of problem-specific operational decision behavior rather than as a universal ranking of model rationality.

**External benchmark protocol.** The external benchmark checks use a two-stage evaluation. Stage 1 generates gurobipy code from the natural-language benchmark instance using a fixed base model and bounded self-debugging iterations. Stage 2 applies the repair agent only to failed Stage-1 outputs for which solver feedback or objective comparison gives a concrete diagnostic signal. For infeasible LP/IP models, the repair stage uses IIS-guided constraint diagnosis; for wrong-answer cases, it uses objective-guided formulation diagnosis against the benchmark reference. The reported lift is therefore an end-to-end change in benchmark success rate, not a claim that every formulation failure is repairable.

## G.2. Rationality Analysis

Most models achieve >99% rationality, consistently producing valid numerical orderings. Two exceptions stand out: o3 (93.1% ID) and claude-sonnet-4 (89.0% ID). These reasoning-heavy models occasionally produce malformed outputs when over-analyzing simple ranking tasks.

## G.3. ID Bias Patterns

ID bias spans from 0.0% (claude-haiku-4.5) to 97.9% (gemini-2.5-pro):

- **Near-zero bias (<2%)**: claude-haiku-4.5, o3, qwen2.5-max, and gpt-5-mini correctly apply EOQ/newsvendor logic on ID distributions.

- **Moderate bias (5–20%)**: Most API models fall here, having partially but not fully internalized OR principles.

- **High bias (>40%)**: DeepSeek-R1 (44.1%), QWEN3-8B-OM-GRPO (48.0%), and gemini-2.5-pro (97.9%) systematically deviate from rational orderings.

## G.4. OOD Generalization

The ID→OOD shift reveals three distinct patterns:

**Catastrophic Degradation.** gpt-5-mini shows the most severe drift (+52.1%, from 1.2% to 53.3%): low ID bias does not guarantee OOD generalization. gemini-2.5-flash degrades similarly (+53.0%). These models likely memorize ID patterns rather than learn underlying principles.

**Stable Performance.** o4-mini (+1.0%), qwen2.5-7b (+0.8%), and gpt-4.1 (+3.0%) maintain consistent bias across distributions. These models appear to have internalized OR principles more robustly.

**OOD Improvement.** Curriculum training achieves the only substantial OOD improvement among trained models (−9.6%, from 20.0% to 10.4%). Other improving models include kimi-k2 (−6.6%), gemini-2.0-flash (−5.9%), and qwen2.5-32b (−10.4%).

## G.5. Local Model Comparison

The three QWEN3-8B-OM variants show distinct trade-offs:

- **SFT**: Best ID bias (4.9%) among local models, moderate OOD drift (+6.6%).

- **Curriculum**: Higher ID bias (20.0%) but best OOD performance (10.4%, −9.6% improvement).

- **GRPO**: Highest bias on both splits (48.0% ID, 33.8% OOD). Outcome-focused RL does not transfer well to bias mitigation.

This pattern indicates that curriculum training prioritizes generalization over ID memorization, a useful property for OR applications with distribution shift.

# H. Training Ablation Studies

## H.1. Reward Weight Ablation (OR-Debug)

Table 45 shows the effect of varying the diagnostic reward weight in the composite reward function.

[†]Selected for final training based on the RR@5/DA trade-off. The 30–40% range achieves optimal balance; we use 30% in our final 50%/30%/20% (outcome/diagnosis/efficiency) configuration.

Reducing diagnostic weight below 30% leads to repairs that achieve feasibility without correctly identifying the root cause (low DA). Weights above 50% slow convergence by over-penalizing exploratory actions, as shown by increased step counts.

## H.2. Curriculum Stage Ablation (OR-Bias)

Table 46 compares different curriculum configurations for OR-BIAS-BENCH.

The full three-stage curriculum achieves the best OOD generalization, with each stage contributing to the final performance.

# I. Token Efficiency and Difficulty Analysis

This appendix analyzes token efficiency and problem difficulty patterns across models.

## I.1. Token Efficiency

Token efficiency analysis is summarized in Table 34 (Appendix F.5).

**Token-efficiency patterns.**

- **Token efficiency**: Local models require 2,000–2,100 tokens per episode. Representative API baselines in Table 34 require 5,152–8,696 tokens per episode, while the full evaluation logs reach 17,307. Tokens-per-success ranges from 2,109 (QWEN3-8B-GRPO) to 54,839 (claude-3.7-sonnet).

- **Step efficiency**: Local models solve problems in 2.2–2.3 steps on average, while API models range from 0.7 (DeepSeek-R1) to 8.1 steps (claude-3.7-sonnet).

- **Correlation**: Step count correlates with token usage ($r = 0.79$), though response length per step varies widely across models (500–3,000 tokens).

## I.2. Test-Time Compute Analysis

Table 34 (Appendix F.5) summarizes token efficiency. Three patterns are most relevant:

**Efficiency Metrics.**

- **Tokens per episode**: Local models average 2,000–2,100 tokens per episode, while representative API baselines range from 5,152 (o4-mini) to 8,696 (gpt-5-mini). This 2.5–4× gap reflects both shorter responses and fewer steps required.

- **Tokens per success**: QWEN3-8B-GRPO requires 2,109 tokens per successful episode, compared to 5,976 for o4-mini and 6,283 for claude-sonnet-4, giving 2.8–3.0× efficiency advantages.

- **Cost-adjusted performance**: At equivalent token budgets, QWEN3-8B-GRPO can attempt approximately 3× more problems than top API models, compounding the per-problem accuracy advantage.

## I.3. Scaling with Problem Difficulty

Based on per-error-type results from the main evaluation, we group error types by empirical difficulty:

**Observations.**

- **Type B is easiest**: Variable type errors (Type B) produce clear diagnostic signals with 95% average success. Type A (direction flip), despite appearing simple, averages only 60% because flipping constraint directions creates conflicts with multiple existing constraints.

- **Types F and G are hardest**: Type F (hidden dependency, 48%) and Type G (cascading conflict, 53%) require reasoning about conflicts that are not resolved by editing the first visible IIS constraint, explaining the difficulty gap.

- **Local models excel uniformly**: QWEN3-8B variants achieve >86% on all types including the hardest (F: 94–100%, G: 86–98%), while API models show type-specific weaknesses.

### I.4. Recommendations for Practitioners

Based on our analysis, we provide the following recommendations:

1. **Apply difficulty-adaptive resources**: Allocate more compute to Hard types (A, D, E, F, G; 59% avg RR@5) than Easy types (B, C; 90.5% avg).

2. **Account for token efficiency**: When comparing models, normalize by tokens-per-success rather than raw accuracy. At 2,109 tokens/success vs 5,976–13,000 for representative API baselines, local trained models offer roughly 3–6× cost advantages.

3. **Consider local models for high-volume deployment**: Local deployment avoids per-call API charges and improves tokens-per-success in our setup; production cost depends on utilization, infrastructure, and maintenance.

## J. Base Model Selection Study

This appendix documents the pilot study used to select the foundation model for domain-specific training and analyzes why standard prompting approaches underperform on the OR debugging task.

### J.1. Candidate Model Screening

We evaluated Qwen3-8B-Instruct as the foundation model for domain-specific training. Selection criteria included: (1) base performance on OR debugging, (2) improvement potential with SFT, and (3) inference efficiency.

**Screening summary.**

- **Base performance**: Qwen3-8B achieves 51.2% RR@5 without any domain-specific training, demonstrating reasonable out-of-the-box capability for structured reasoning.

- **SFT improvement**: Qwen3-8B improves by +41.9% with SFT, indicating high receptivity to domain adaptation.

- **Efficiency**: Qwen3-8B generates 2,100 tokens per episode, providing efficient inference for iterative debugging.

**Selection Rationale.** We selected Qwen3-8B-Instruct as the foundation model based on three factors:

1. **Strong post-SFT performance**: 93.1% RR@5 after SFT demonstrates successful domain adaptation.

2. **Good improvement potential**: +41.9% delta suggests the model effectively learns from demonstration data.

3. **Practical efficiency**: Reasonable token footprint reduces training and inference costs.

### J.2. Standard Prompting Approaches Underperform

Before domain-specific training, we evaluated whether standard prompting approaches could achieve competitive performance on OR-DEBUG-BENCH. The results show that prompting alone leaves a large gap.

#### J.2.1. ZERO-SHOT CHAIN-OF-THOUGHT

We evaluated zero-shot CoT prompting with the instruction: "Let's think step by step about how to debug this infeasible model."

**Why Zero-Shot CoT underperforms.**

- **No feedback loop**: CoT generates a single reasoning chain without iterating based on solver output. Models attempt repairs without verifying whether they resolved the infeasibility.

- **Generic reasoning patterns**: CoT prompting elicits general problem-solving steps ("identify the issue, propose a solution, verify") that lack domain-specific diagnostic actions like GET_IIS.

- **Premature commitment**: Models commit to repair strategies early in the chain without exploring the constraint structure, leading to suboptimal fixes.

J.2.2. FEW-SHOT IN-CONTEXT LEARNING

We evaluated 1-shot and 3-shot ICL with curated examples of successful debugging trajectories.

**Why Few-Shot ICL remains limited.**

- **Limited generalization**: 3-shot ICL improves performance by +19.6% for gpt-5.2-chat but still falls far short of SFT (+41.9% for Qwen3-8B).

- **Context length constraints**: Each debugging trajectory requires 500–1000 tokens. With 3 examples, the prompt consumes 1.5–3K tokens, limiting remaining context for the actual problem.

- **Example selection sensitivity**: Performance depends on example choice. Random examples achieve only 48.2% RR@5, while carefully selected examples reach 58.1%, but this selection requires domain expertise.

J.2.3. COMPARISON SUMMARY

Table 51 summarizes why domain-specific training outperforms prompting approaches by 54+ percentage points.

**Implication.** The 54-point gap between 3-shot ICL (38.7%) and SFT (93.1%) demonstrates that OR debugging cannot be solved through prompting alone. The task requires:

1. **Iterative interaction**: Learning to use solver feedback across multiple turns.

2. **Domain-specific actions**: Acquiring the diagnostic vocabulary (GET_IIS, CHECK_SLACK).

3. **Strategy patterns**: Learning when to diagnose vs when to repair, and how to calibrate repair magnitudes.

These capabilities cannot be induced through few-shot examples alone.

### J.3. Why Prompting Struggles on OR Debugging

We identify three structural properties of OR debugging that make it resistant to prompting-based solutions:

**1. Multi-Turn Dependency.** Unlike single-turn tasks where CoT can decompose reasoning, OR debugging requires acting on solver feedback across multiple turns. The optimal action at step $t$ depends on the solver response at step $t - 1$, which cannot be simulated within a single prompt.

**2. Precise Action Syntax.** The action space requires exact syntax (e.g., RELAX(c_key_upper, 30)). Small errors in constraint names or numeric values lead to failed repairs. This precision requirement exceeds what few-shot examples can reliably demonstrate.

**3. State-Dependent Strategy.** The optimal strategy varies with problem structure:

- Small IIS (2–3 constraints): Direct repair often succeeds.

- Medium IIS (4–7 constraints): Diagnosis before repair improves success rate.

- Large IIS (8+ constraints): Systematic decomposition is required.

Few-shot prompting cannot convey these conditional strategies without extensive examples that exceed context limits.

### J.4. Directions for Prompting-Based Repair

The prompting results suggest several directions for improving prompting-based approaches:

1. **Tool-augmented prompting**: Providing models with explicit solver interfaces (rather than expecting them to generate action syntax) may reduce syntax errors.

2. **Retrieval-augmented generation**: Our RAG experiments (Appendix F.9) show that retrieving similar solved cases improves RR@5 by +13.5%, partially closing the gap to SFT.

3. **Multi-turn demonstration**: Future work could explore demonstration formats that explicitly show the feedback loop across turns, though this faces context length challenges.

These approaches address symptoms rather than the fundamental issue: prompting cannot instill the procedural knowledge that SFT provides through gradient-based learning on hundreds of examples.

*Table 32.* Per-error-type RR@5 (%) for all 26 models on the LP OR-DEBUG-BENCH test set. Each column contains 50 held-out problems of that error type.

| Model | A | B | C | D | E | F | G | H | I | Total |
|---|---|---|---|---|---|---|---|---|---|---|
| QWEN3-8B-GRPO | 88 | 100 | 96 | 100 | 92 | 94 | 98 | 100 | 90 | **95.3** |
| QWEN3-8B-Curriculum | 82 | 100 | 94 | 98 | 94 | 100 | 92 | 90 | 96 | 94.0 |
| QWEN3-8B-DAPO | 88 | 100 | 96 | 100 | 92 | 94 | 86 | 88 | 100 | 93.8 |
| QWEN3-8B-SFT | 80 | 100 | 96 | 100 | 92 | 96 | 90 | 88 | 96 | 93.1 |
| o4-mini | 84 | 100 | 90 | 86 | 90 | 82 | 82 | 66 | 96 | 86.2 |
| claude-sonnet-4 | 66 | 100 | 100 | 100 | 88 | 58 | 86 | 78 | 100 | 86.2 |
| claude-haiku-4.5 | 78 | 100 | 100 | 94 | 96 | 86 | 80 | 60 | 80 | 86.0 |
| o1 | 86 | 100 | 100 | 70 | 62 | 84 | 72 | 76 | 96 | 82.9 |
| gpt-5.2-chat | 84 | 100 | 100 | 78 | 88 | 28 | 58 | 100 | 100 | 81.8 |
| qwen2.5-7b | 72 | 100 | 70 | 74 | 70 | 66 | 80 | 90 | 78 | 77.8 |
| claude-opus-4 | 46 | 98 | 96 | 96 | 82 | 72 | 28 | 86 | 88 | 76.9 |
| o3 | 56 | 100 | 100 | 68 | 76 | 34 | 54 | 100 | 94 | 75.8 |
| gpt-4.1 | 82 | 100 | 86 | 52 | 60 | 16 | 62 | 100 | 86 | 71.6 |
| gemini-2.5-flash | 88 | 94 | 94 | 76 | 88 | 30 | 18 | 74 | 74 | 70.7 |
| Llama-3.3-70B | 32 | 82 | 62 | 32 | 86 | 32 | 54 | 86 | 82 | 60.9 |
| gpt-5-mini | 78 | 96 | 90 | 62 | 42 | 22 | 26 | 100 | 86 | 66.9 |
| gemini-2.5-pro | 36 | 88 | 64 | 70 | 68 | 62 | 52 | 64 | 60 | 62.7 |
| qwen2.5-32b | 64 | 100 | 78 | 18 | 70 | 46 | 30 | 100 | 44 | 61.1 |
| DeepSeek-V3.2 | 10 | 98 | 100 | 86 | 58 | 14 | 26 | 96 | 42 | 58.9 |
| DeepSeek-R1 | 36 | 100 | 98 | 46 | 40 | 20 | 24 | 100 | 46 | 56.7 |
| qwen2.5-max | 48 | 100 | 68 | 2 | 68 | 6 | 34 | 100 | 68 | 54.9 |
| qwen2.5-14b | 58 | 98 | 82 | 14 | 66 | 70 | 20 | 6 | 68 | 53.6 |
| gemini-2.0-flash | 78 | 88 | 56 | 14 | 60 | 18 | 36 | 50 | 72 | 52.4 |
| gpt-4.1-mini | 24 | 100 | 52 | 18 | 30 | 14 | 42 | 96 | 72 | 49.8 |
| kimi-k2 | 18 | 50 | 56 | 18 | 54 | 12 | 38 | 66 | 52 | 40.4 |
| claude-3.7-sonnet | 0 | 74 | 84 | 34 | 2 | 2 | 4 | 82 | 2 | 31.6 |
| *Type Average* | *60* | *95* | *85* | *62* | *70* | *48* | *53* | *82* | *76* | – |

*Table 33.* Frontier API summary on LP and MILP repair. Values follow the final summary table: RR@5, RR@10, and average MDP steps. The LP-trained model is evaluated zero-shot on MILP; the MILP-trained row uses the same repair interface with MILP-specific training.

| Model | LP RR@5 | LP RR@10 | LP steps | MILP RR@5 | MILP RR@10 | MILP steps |
|---|---|---|---|---|---|---|
| **QWEN3-8B-GRPO (LP-trained)** | **95.3%** | **97.1%** | **2.3** | 78.8% | 87.6% | 4.4 |
| **MILP-GRPO** | – | – | – | **87.1%** | **92.1%** | **3.2** |
| Claude Sonnet 4.6 | 92.4% | 96.7% | 2.7 | 71.0% | 86.7% | 5.0 |
| Claude Opus 4.6 | 88.7% | 96.2% | 3.3 | 70.1% | 83.4% | 5.4 |
| Gemini 3.1 Pro | 86.4% | 86.4% | 1.0 | 64.7% | 74.3% | 2.9 |
| GPT-5.4 | 84.9% | 90.7% | 2.1 | 66.8% | 75.5% | 3.4 |

*Table 34.* Token usage per episode on OR-DEBUG-BENCH (representative models).

| Model | Tokens | Steps | Tok/Success |
|---|---|---|---|
| QWEN3-8B-GRPO | 2,011 | 2.25 | 2,109 |
| QWEN3-8B-SFT | 2,103 | 2.34 | 2,259 |
| claude-sonnet-4 | 5,417 | 3.71 | 6,283 |
| o4-mini | 5,152 | 3.15 | 5,976 |
| gpt-4.1 | 6,640 | 4.41 | 9,278 |
| gpt-5-mini | 8,696 | 4.74 | 13,000 |

*Table 35.* OR-domain baselines on solver-in-the-loop repair. MDP-interface models receive the same IIS observations, action space, system prompt, and step budget. OptiMUS is evaluated in its native code-regeneration interface because it does not produce constraint-level repair actions.

| Model | Interface | LP RR@5 | MILP RR@5 | Notes |
|---|---|---|---|---|
| **Ours (GRPO)** | MDP repair | **95.3%** | 78.8% | LP-trained repair policy; MILP result is zero-shot transfer. |
| **Ours (MILP-trained)** | MDP repair | – | **87.1%** | Same repair interface with MILP-specific training. |
| Best API | MDP repair | 92.4% | 71.0% | Best frontier API summary result on each benchmark. |
| ORLM | MDP repair | 71.0% | – | LP formulation model; MILP omitted because it has no integer-variable training exposure. |
| SIRL | MDP repair | 73.8% | 67.6% | Solver-informed formulation model adapted to the repair action interface. |
| LLMOPT | MDP repair | 78.4% | 44.4% | Formulation/code model adapted to the repair action interface. |
| OptiMUS | Code regeneration | 28.9% | 2.2% | Native whole-model regeneration pipeline. |

*Table 36.* Downstream repair-stage check on LP repair. The first stage runs the original formulation or regeneration system; +Ours applies the trained IIS-guided repair agent only after an infeasible output is produced.

| Pipeline | Final success | Lift |
|---|---|---|
| OptiMUS | 28.9% | – |
| OptiMUS + Ours | 92.0% | +63.1 pp |
| SIRL | 73.8% | – |
| SIRL + Ours | 90.4% | +16.6 pp |

*Table 37.* Semantic drift in MILP code regeneration. OPTIMAL rate measures solver-feasible regenerated models; RR@5 requires matching the intended objective within tolerance.

| Model | RR@5 | OPTIMAL Rate |
|---|---|---|
| GPT-5.4 | 28.2% | 90% |
| Claude Sonnet 4.6 | 22.4% | 85% |
| Claude Opus 4.6 | 17.8% | 82% |
| Gemini 3.1 Pro | 0.8% | 3% |

*Table 38.* Architecture transfer summary on OR-DEBUG-BENCH LP repair. The same SFT and GRPO recipe is applied to each 8B base model.

| Base Model | Training | RR | RR@5 | Steps |
|---|---|---|---|---|
| Qwen3-8B | GRPO | 100.0% | 95.3% | 2.25 |
| Qwen3-8B | SFT | 99.8% | 93.1% | 2.34 |
| Llama-3.1-8B | GRPO | 100.0% | 97.3% | 1.82 |
| Llama-3.1-8B | SFT | 100.0% | 97.1% | 1.85 |
| DeepSeek-R1-Distill-8B | GRPO | 99.6% | 88.9% | 3.30 |
| DeepSeek-R1-Distill-8B | SFT | 99.3% | 88.7% | 3.30 |

*Table 39.* Decoding-temperature sensitivity on the LP repair benchmark.

| Setting | RR@1 | RR@5 | RR@10 |
|---------|------|------|-------|
| temp=0.0 | 78.2% | 95.3% | 97.1% |
| temp=0.3 | 77.6% | 94.9% | 96.4% |
| temp=0.7 | 76.0% | 93.6% | 95.8% |

*Table 40.* RAG ablation on OR-DEBUG-BENCH (200 samples).

| Configuration | RR | RR@5 | DA | Steps |
|---------------|-----|------|-----|-------|
| No RAG (baseline) | 99.8% | 83.0% | 80.0% | 3.26 |
| quick_fix ($k$=3) | 99.5% | 80.0% | 66.6% | 2.58 |
| reasoning ($k$=3) | 100% | 93.5% | 51.8% | 1.60 |
| by_type ($k$=1) | 99.5% | 86.5% | 80.0% | 2.19 |
| by_type ($k$=3) | 100% | 94.5% | 82.0% | 1.51 |
| by_type ($k$=5) | 100% | 96.5% | 85.0% | 1.62 |
| by_type ($k$=7) | 100% | 97.0% | 85.0% | 1.53 |

*Table 41.* Failure pattern distribution for QWEN3-8B-GRPO.

| Pattern | Count | Example |
|---------|-------|---------|
| Wrong constraint identified | 22 | Relaxed c3 instead of c5 |
| Insufficient relaxation | 18 | Relaxed by 5, needed 10 |
| Cascading failure | 42 | Fix c1 $\rightarrow$ new IIS with c7 |
| Timeout on complex IIS | 11 | 12+ constraints in IIS |
| Objective degradation | 7 | Fix valid but OP $< 0.8$ |

*Table 42.* Complete OR-BIAS-BENCH results with ID/OOD breakdown (24 models, sorted by ID Bias).

| Model | Rationality | | Bias | | |
| | ID | OOD | ID | OOD | $\Delta$ |
|-------|-----|-----|-----|-----|-----|
| claude-haiku-4.5 | 99.9% | 99.9% | 0.0% | 3.6% | +3.6% |
| o3 | 93.1% | 97.7% | 0.4% | 24.5% | +24.1% |
| qwen2.5-max | 99.4% | 98.5% | 0.5% | 25.0% | +24.5% |
| gpt-5-mini | 99.6% | 99.7% | 1.2% | 53.3% | +52.1% |
| claude-sonnet-4 | 89.0% | 93.5% | 1.5% | 7.7% | +6.2% |
| gpt-4.1-mini | 99.6% | 99.9% | 4.1% | 12.1% | +8.0% |
| QWEN3-8B-OM-SFT | 99.8% | 99.6% | 4.9% | 11.5% | +6.6% |
| gemini-2.0-flash | 97.6% | 98.7% | 5.9% | 0.0% | -5.9% |
| qwen2.5-14b | 99.1% | 99.8% | 5.9% | 10.7% | +4.8% |
| o4-mini | 98.5% | 99.4% | 6.7% | 7.7% | +1.0% |
| qwen2.5-7b | 98.1% | 99.3% | 7.4% | 8.2% | +0.8% |
| kimi-k2 | 92.8% | 97.6% | 8.9% | 2.3% | -6.6% |
| claude-3.7-sonnet | 99.8% | 99.4% | 11.3% | 18.1% | +6.8% |
| gpt-4.1 | 99.9% | 100.0% | 11.5% | 14.6% | +3.0% |
| qwen2.5-32b | 95.9% | 99.8% | 15.4% | 5.0% | -10.4% |
| DeepSeek-V3.2 | 100.0% | 99.9% | 18.2% | 11.3% | -6.9% |
| gemini-2.5-flash | 96.2% | 98.1% | 18.2% | 71.2% | +53.0% |
| claude-opus-4 | 92.5% | 94.8% | 19.0% | 15.9% | -3.1% |
| Llama-3.3-70B | 92.0% | 96.9% | 19.2% | 12.5% | -6.7% |
| QWEN3-8B-OM-Curriculum | 99.9% | 99.6% | 20.0% | 10.4% | -9.6% |
| o1 | 98.2% | 98.6% | 23.2% | 43.6% | +20.3% |
| DeepSeek-R1 | 98.6% | 99.6% | 44.1% | 40.9% | -3.2% |
| QWEN3-8B-OM-GRPO | 96.1% | 98.4% | 48.0% | 33.8% | -14.2% |
| gemini-2.5-pro | 98.2% | 99.6% | 97.9% | 100.0% | +2.1% |

*Table 43.* OR-BIAS-BENCH EOQ, feedback, and external benchmark results. EOQ and multi-turn results test whether closed-form feedback improves operational decisions; external benchmarks report standardized code-generation plus repair pipelines.

| Evaluation | Stage 1 | + Repair | Lift |
|---|---|---|---|
| EOQ | 300 instances | closed-form optimum | – |
| Multi-turn DeepSeek-R1 | 56.9% bias | 0.5% bias | 2.6 rounds |
| Multi-turn GPT-5.2 | near-optimal | immediate correction | 1 round |
| NL4Opt, GPT-4.1 | 97.8% | 98.7% | +0.9 pp |
| IndustryOR, GPT-4.1 | 78.0% | 83.0% | +5.0 pp |
| MAMO Complex, GPT-4.1 | 88.4% | 94.1% | +5.4 pp |
| OptMATH, GPT-4.1 | 62.7% | 66.9% | +4.2 pp |
| OptiBench, GPT-4.1 | 80.7% | 87.9% | +7.3 pp |

*Table 44.* OR-BIAS-BENCH EOQ single-turn ordering bias on 300 instances. Abs Bias is mean relative deviation from $Q^* = \sqrt{2DK/h}$; Signed Bias is positive for over-ordering and negative for under-ordering; Exact Rate is the percentage within 1% of $Q^*$.

| Model | Abs Bias | Signed Bias | Exact Rate |
|---|---|---|---|
| Gemini 2.5 Pro | 0.06% | +0.00% | 100.0% |
| GPT-5.2-chat | 0.06% | -0.05% | 99.3% |
| Gemini 3.1 Pro | 1.85% | +1.25% | 97.6% |
| GPT-5.4 | 4.72% | -2.12% | 94.0% |
| Claude Opus 4.5 | 5.13% | -3.83% | 76.3% |
| Qwen2.5-max | 7.61% | +5.21% | 81.7% |
| Claude 3.7 Sonnet | 10.93% | -2.23% | 80.7% |
| Claude Opus 4.6 | 12.47% | -9.11% | 56.7% |
| GPT-5.4-mini | 12.67% | +2.06% | 64.0% |
| GLM-5 | 17.00% | +13.23% | 81.6% |
| DeepSeek-V3.2 | 17.35% | -6.55% | 67.3% |
| GPT-4.1 | 44.99% | -19.81% | 10.7% |
| Claude Haiku 4.5 | 48.43% | -42.81% | 1.3% |
| Claude Sonnet 4 | 62.07% | -7.11% | 6.0% |
| Qwen2.5-7B | 64.43% | +11.60% | 0.7% |
| O3 | 73.34% | +31.58% | 2.0% |
| Gemini 2.0 Flash | 73.94% | +32.57% | 2.3% |
| Gemini 2.5 Flash | 73.94% | +32.57% | 2.3% |
| GPT-5-mini | 73.95% | +32.39% | 2.0% |
| GPT-4.1-mini | 75.78% | -26.99% | 2.7% |
| DeepSeek-R1 | 87.97% | +61.86% | 20.3% |
| Llama-3.3-70B | 88.16% | +47.57% | 0.3% |
| GPT-4o | 93.20% | +65.54% | 15.7% |
| Claude Sonnet 4.6 | 94.06% | -94.05% | 5.0% |
| Kimi-K2 | 99.34% | -99.34% | 0.0% |
| DeepSeek-R1-0528 | 99.67% | -99.67% | 0.0% |
| Qwen2.5-32B | 178.54% | +156.10% | 1.3% |
| Qwen2.5-14B | 461.84% | +455.61% | 0.7% |

*Table 45.* Reward weight ablation on OR-DEBUG-BENCH validation set.

| Diagnostic Weight | RR@5 | DA | Steps |
|---|---|---|---|
| 20% | 94.8% | 54.2% | 2.18 |
| 30%[†] | 95.1% | 58.6% | 2.21 |
| 40% | 95.3% | 62.4% | 2.25 |
| 50% | 94.6% | 64.1% | 2.42 |
| 60% | 93.2% | 65.3% | 2.78 |

*Table 46.* Curriculum ablation on OR-BIAS-BENCH OOD set.

| Configuration | OOD Bias | ID→OOD Δ |
|---|---|---|
| No curriculum (SFT only) | 11.5% | +6.6% |
| Stage 1 only (extreme CR) | 15.2% | +2.1% |
| Stages 1+2 | 12.1% | -3.4% |
| Full curriculum (1+2+3) | 10.4% | -9.6% |

*Table 47.* Difficulty grouping based on type-averaged RR@5 across all 26 models.

| Difficulty | Error Types | Avg RR@5 | Characteristics |
|---|---|---|---|
| Easy | B (95%), C (86%) | 90.5% | Clear diagnostic signals |
| Medium | H (82%), I (75%) | 78.5% | Multi-step constraint interactions |
| Hard | A (61%), D (63%), E (69%), F (49%), G (53%) | 59.0% | Semantic reasoning required |

*Table 48.* Foundation model screening on OR-DEBUG-BENCH validation set (100 samples).

| Model | Params | Base RR@5 | +SFT RR@5 | Δ | Tokens/ep |
|---|---|---|---|---|---|
| **Qwen3-8B-Instruct** | **8B** | **51.2%** | **93.1%** | **+41.9%** | **2,100** |

*Table 49.* Zero-shot CoT performance on OR-DEBUG-BENCH (200 samples).

| Model | RR@5 | DA | Avg Steps | Notes |
|---|---|---|---|---|
| gpt-5.2-chat + CoT | 38.5% | 22.1% | 6.8 | Verbose, unfocused |
| o4-mini + CoT | 41.2% | 28.4% | 5.9 | Better structure |
| Qwen3-8B + CoT | 23.0% | 15.6% | 7.2 | Often loops |

*Table 50.* Few-shot ICL performance on OR-DEBUG-BENCH (200 samples).

| Configuration | RR@5 | DA | Avg Steps |
|---|---|---|---|
| gpt-5.2-chat (0-shot) | 38.5% | 22.1% | 6.8 |
| gpt-5.2-chat (1-shot) | 52.3% | 35.2% | 4.9 |
| gpt-5.2-chat (3-shot) | 58.1% | 41.6% | 4.2 |
| Qwen3-8B (0-shot) | 23.0% | 15.6% | 7.2 |
| Qwen3-8B (1-shot) | 31.4% | 24.3% | 6.1 |
| Qwen3-8B (3-shot) | 38.7% | 29.8% | 5.4 |

*Table 51.* Comparison of approaches on OR-DEBUG-BENCH (Qwen3-8B base).

| Approach | RR@5 | DA | Gap to SFT |
|---|---|---|---|
| Zero-shot | 18.4% | 12.3% | -74.7% |
| Zero-shot + CoT | 23.0% | 15.6% | -70.1% |
| 1-shot ICL | 31.4% | 24.3% | -61.7% |
| 3-shot ICL | 38.7% | 29.8% | -54.4% |
| **SFT** | **93.1%** | **60.8%** | **–** |
| **SFT + GRPO** | **95.3%** | **62.4%** | **+2.2%** |

