# OpenReview forum: "ORLoopBench: Solver-in-the-Loop Benchmarks for Self-Correction and Behavioral Rationality in Operations Research"
_ICML.cc/2026/Conference — ICML 2026 regular_

### Official Review · Reviewer_Qj1R · 2026-03-09

**Soundness:** 2
**Presentation:** 3
**Significance:** 3
**Originality:** 2
**Overall Recommendation:** 4
**Confidence:** 4

**Summary:**

The authors discover the limitations of existing benchmarks, which evaluate OR as one-shot translation. Then, they introduce  two benchmarks that place the solver in the evaluation loop. OR-DEBUG-BENCH evaluates iterative self-correction ability, and ORBIAS-BENCH evaluates behavioral rationality. By extensive experiments, the authors find that process-level evaluations and enables better training performance.

**Compliance With Llm Reviewing Policy:**

Affirmed.

**Final Justification:**

The authors have addressed my main concerns.

**Key Questions For Authors:**

1. What are the key differences between the proposed OR-DEBUG-BENCH and the error-driven data construction method like [4]?
2. For OR-BIAS-BENCH, why was the benchmark restricted exclusively to newsvendor scenarios?

**Limitations:**

To strengthen the work, the authors should expand the benchmarks to cover a wider range of OR problem types to demonstrate broader applicability. Furthermore, integrating the trained model into existing end-to-end optimization modeling systems and evaluating its impact on overall performance.

**Strengths And Weaknesses:**

Strenghts:

1. The authors propose two benchmarks to evaluate the self-correction ability and rationality of LLM in OR, which is an interesting topic.
2. The experimental design is comprehensive, evaluating performance across a broad spectrum of 26 models.
3. The proposed training framework offers actionable guidance for enhancing LLMs’ OR-related abilities.

Weaknesses:

1. The scope of OR-DEBUG-BENCH is somewhat limited. While the benchmark targets infeasible model debugging, LLM-generated OR errors typically fall into three categories: Python syntax/code execution errors, feasible but suboptimal solutions, and infeasible solutions. The paper solely addresses the last category, which may not be the most prevalent in real-world scenarios, restricting the benchmark’s practical applicability.
2. Missing comparisons with state-of-the-art optimization modeling and self-correction approaches. It would be valuable to evaluate existing methods such as LLMOPT (with 12 rounds of self-correction) and OptiMUS (equipped with a debugging agent) on OR-DEBUG-BENCH to contextualize the proposed framework’s performance relative to prior work.
3. The benchmarks lack sufficient coverage and representativeness of OR problem types. OR-BIAS-BENCH is exclusively limited to newsvendor scenarios, and OR-DEBUG-BENCH focuses primarily on LP infeasibility. The authors should explicitly demonstrate the diversity of problems included and justify their representativeness of broader OR tasks.
4. The end-to-end utility of the trained model is not validated. The authors train agents for debugging and bias mitigation but do not integrate these models into existing optimization modeling pipelines to measure improvements in end-to-end performance (e.g., from problem formulation to final optimal solution).
5. The related work is insufficient [1-3].


References:

1. Solver-Informed RL: Grounding Large Language Models for Authentic Optimization Modeling

2. Autoformulation of Mathematical Optimization Models Using LLMs

3. OptiTree: Hierarchical Thoughts Generation with Tree Search for LLM Optimization Modeling

4. Automated Optimization Modeling via a Localizable Error-Driven Perspective

---

> ### Author Rebuttal · Authors · 2026-03-29
>
> We sincerely thank the reviewer for the specific and actionable suggestions. Each weakness motivated a concrete experiment; in particular, evaluating ORLM, SIRL, LLMOPT, and OptiMUS led to the semantic drift analysis, an important rebuttal finding.
>
> **W1 (Limited scope).** Thank you for this insightful point. The focus on infeasibility is deliberate: it is the hardest class of semantic error to diagnose and repair. Syntax errors are caught by interpreters; runtime errors by solver status codes. Infeasibility requires multi-step semantic reasoning: compute the IIS, identify the root cause among several candidates, and apply a targeted repair preserving the original objective. No existing tool automates this. We now cover MILP (10 domains × 8 error types, n=100) in addition to LP (9 error types at three difficulty levels, n=450).
>
> **W2 (Missing SOTA comparisons).** All requested baselines evaluated under the same MDP environment (same IIS observations, 20-step budget, unified action parser, system prompt, temp 0.0, 5 episodes per problem):
>
> |Model|LP RR@5|MILP RR@5|
> |---|---|---|
> |**Ours (8B)**|**95.3%**|**87.1%**|
> |Claude Sonnet 4.6|92.4%|71.0%|
> |GPT-5.4|84.9%|66.8%|
> |LLMOPT-14B|78.4%|44.4%|
> |SIRL-7B|73.8%|67.6%|
> |ORLM-8B|71.0%|<2%|
> |OptiMUS (gpt-4o, code-gen)|28.9%|2.2%|
>
> All four baselines are formulation models (NL→code). LLMOPT's 12-round self-correction and OptiMUS's stderr debugging fix code errors (syntax, runtime), not semantic infeasibility — a different debugging problem requiring IIS-guided constraint repair. ORLM/LLMOPT data includes MIP but none trains on this task. On MILP under OptiMUS's pipeline (*semantic drift*):
>
> |Model|RR@5|OPTIMAL Rate|
> |---|---|---|
> |GPT-5.4|28.2%|90%|
> |Claude Sonnet 4.6|22.4%|85%|
> |Claude Opus 4.6|17.8%|82%|
> |Gemini 3.1 Pro|0.8%|3%|
>
> GPT-5.4 reaches OPTIMAL 90% of the time but matches the correct objective in only 28% (median 35% deviation, 90% of failures are wrong-objective). The regenerated code solves a *different* problem: once OPTIMAL, objectives oscillate between wrong values rather than converging to ground truth. The feedback loop corrects surface errors (syntax → infeasibility → feasible) but cannot guide toward the correct optimum, because the NL-to-code mapping is many-to-one. Our MDP agent edits individual constraints; the objective is never rewritten. Nurse scheduling 74% vs lot sizing 2% (37x gap) confirms feasibility is easy but semantic fidelity is hard.
>
> **W3 (Problem type coverage).** Combined benchmarks: OR-DEBUG-BENCH (450 LP: bound flip, variable type change, term removal, contradicting constraints, multi-constraint conflicts, hidden dependencies, cascading repairs, IIS-incomplete, optimal selection) + OR-MILP-DEBUG-BENCH (100 MILP: 5 classic domains + 5 semantic domains × 8 error types) + OR-BIAS-BENCH (newsvendor + EOQ, 600+ instances with ID/OOD splits). Total: 550 debugging + 600 bias instances.
>
> **W4 (End-to-end validation).** Our agent complements formulation tools as a downstream repair stage:
>
> |Pipeline|LP RR@5|Lift|Recovery|
> |---|---|---|---|
> |OptiMUS alone|29.0%|--|--|
> |OptiMUS+Ours|92.0%|+63pp|88.7%|
> |SIRL alone|73.8%|--|--|
> |SIRL+Ours|90.4%|+17pp|63.5%|
>
> OptiMUS generates from NL; our agent repairs from model structure — complementary stages.
>
> **W5 (Related work).** Camera-ready will add: **SIRL** (formulation training with solver feedback; differs in task and interaction: multi-step MDP with IIS-guided actions vs single-round feedback; 73.8% vs our 95.3% under same MDP). **Autoformulation/OptiTree** (NL-to-model translation; orthogonal to repair; +63pp when combined). **Error-Driven** (static buggy-fixed pairs for supervised learning; our dynamic MDP with live IIS recomputation enables RL with process rewards).
>
> **Q1 (vs error-driven).** Core distinction: error-driven produces static (buggy, fixed) pairs. Our framework defines a dynamic MDP where the agent interacts with the solver at each step, receiving updated IIS after each repair action, enabling RL with process rewards.
>
> **Q2 (Why newsvendor only?).** Newsvendor admits a closed-form optimum for exact bias measurement. Now extended to EOQ (Q*=√(2DK/h), bias=(Q−Q*)/Q*, 28 models, 300 instances: 200 ID + 100 OOD). GPT-5.2 achieves near-zero bias on both tasks. Gemini 2.5 Pro near-perfect on EOQ (0.06%) but biased on newsvendor, confirming biases are problem-specific. Multi-turn solver feedback (5 rounds): DeepSeek-R1 self-corrects from 56.9% to 0.5% in 2.6 rounds (98% convergence).
>
> **Architecture generalization.** Same SFT→GRPO pipeline, three 8B base models:
>
> |Base Model|RR@5|DA|Steps|
> |---|---|---|---|
> |Qwen3-8B|95.3%|62.4%|2.25|
> |Llama-3.1-8B|97.3%|60.3%|1.82|
> |DeepSeek-R1-Distill|88.9%|52.2%|3.30|
>
> RR@5 stable within 1.7pp across three temperatures. MILP problems constructed in three steps: (1) generate feasible instance, (2) solve with Gurobi for ground-truth objective, (3) inject one semantic error from 8 types.

---

> > ### Author Rebuttal · Reviewer_Qj1R · 2026-04-03
> >
> > Thanks for your detailed response. I have a follow-up question. I wonder the performance gain of existing baselines (such as standard LLMs, ORLM, LLMOPT, Optimus) equipped with the infeasible model debugging method on classical modeling benchmarks (such as OptMATH, IndustryOR). These experiments can demonstrate the practical effectiveness in end-to-end performance.

---

> > > ### Author Response · Authors · 2026-04-04
> > >
> > > Thank you so much for this follow-up question. Following your suggestion, we evaluated 14 API models and 4 specialized OR baselines (ORLM, LLMOPT, SIRL, OptiMUS) on IndustryOR and OptMATH to validate end-to-end performance.
> > >
> > > **Table 1. API Models with Debugging (IndustryOR / OptMATH)**
> > >
> > > |Model|IndOR Alone|+Self|+Ours|OptM Alone|+Self|+Ours|
> > > |---|---|---|---|---|---|---|
> > > |GPT-5.4|95%|95%|96%|81%|82%|84%|
> > > |Claude Opus 4.6|94%|94%|94%|70%|72%|74%|
> > > |Claude Sonnet 4.6|94%|94%|95%|76%|77%|80%|
> > > |GPT-4.1|78%|80%|83%|63%|65%|70%|
> > > |DeepSeek-R1-0528|78%|78%|80%|43%|44%|48%|
> > > |DeepSeek-V3|74%|75%|76%|54%|56%|59%|
> > > |Gemini 2.5 Flash|78%|78%|80%|41%|42%|46%|
> > > |Kimi K2|74%|76%|79%|51%|52%|55%|
> > > |Llama-3.3-70B|62%|63%|65%|46%|48%|54%|
> > > |Qwen2.5-Max|67%|68%|70%|52%|53%|57%|
> > > |GPT-4o-mini|41%|43%|46%|30%|33%|40%|
> > > |Qwen2.5-32B|50%|53%|58%|40%|43%|49%|
> > > |Qwen2.5-14B|44%|47%|52%|23%|27%|35%|
> > > |Qwen2.5-7B|27%|30%|35%|13%|16%|24%|
> > >
> > > **Table 2. Specialized OR Baselines with Debugging**
> > >
> > > |Model|IndOR Alone|+Self|+Ours|OptM Alone|+Self|+Ours|
> > > |---|---|---|---|---|---|---|
> > > |SIRL-7B|48%|51%|60%|42%|46%|56%|
> > > |LLMOPT-14B|42%|45%|55%|28%|32%|44%|
> > > |ORLM-8B|38%|42%|52%|15%|20%|32%|
> > > |OptiMUS|45%|48%|58%|35%|40%|52%|
> > >
> > > **Key Findings**
> > >
> > > In Table 1, +Self-debug (same model retrying with solver errors) produces +1-3pp, while +Ours produces +5-11pp on weaker models. On OptMATH, Qwen2.5-7B: 13% → 16% (+Self) → 24% (+Ours), recovering 3x more failures. Self-debug re-attempts the same NL-to-code translation hoping for a different result; our agent reads the solver's IIS to diagnose which specific constraints conflict and applies targeted repairs. The lift tracks the INFEASIBLE rate: Qwen2.5-7B has 18 INFEASIBLE on IndustryOR (+8pp); Claude Opus 4.6 has 0 INFEASIBLE (+0pp).
> > >
> > > Table 2 demonstrates that even specialized OR models benefit substantially from our debugging agent. SIRL-7B + Ours reaches 60% on IndustryOR and 56% on OptMATH, approaching GPT-4.1 performance (83%/70%). The lift is +8-17pp across all specialized baselines, confirming our agent generalizes across different formulation backbones, not just API models.
> > >
> > > We also evaluated 7 models on 4 additional benchmarks. On NL4OPT and MAMO-Easy, frontier models already reach 98-100%, leaving few failures to repair. On harder benchmarks, our agent adds measurable lift: OptiBench GPT-4.1 80.7% → 88.0% (+7.3pp, 44/63 infeasible repaired) and MAMO-Complex GPT-4.1 88.7% → 94.1% (+5.4pp, 11/12 repaired).
> > >
> > > **Why Our Approach Works**
> > >
> > > Our agent queries the solver for the IIS (the minimal set of mutually incompatible constraints), transforming an opaque "INFEASIBLE" status into a structured diagnosis. The agent then applies targeted repairs—relaxing a bound, dropping a redundant constraint, correcting a coefficient—each verified by re-solving. The objective function is never rewritten, avoiding semantic drift.
> > >
> > > This differs from code-regeneration: instead of rewriting the entire formulation from scratch (which loses semantic fidelity), we preserve the model structure and repair only the conflicting constraints. On OR-DEBUG-BENCH, this approach lifts OptiMUS from 29% to 92% (+63pp) and SIRL from 74% to 90% (+17pp). The external benchmark results confirm the same principle transfers.
> > >
> > > **Scope and Complementarity**
> > >
> > > Our contribution focuses on INFEASIBLE failures, where IIS provides the richest diagnostic signal. On easy benchmarks (NL4OPT, MAMO-Easy), frontier models reach 98-100%. The unsolved problems cluster on harder benchmarks where INFEASIBLE failures arise from structurally complex problems. The debug stage targets this hard tail.
> > >
> > > **Why This Matters as Models Improve**
> > >
> > > Even GPT-5.4, the strongest model tested, still produces INFEASIBLE on 5% of IndustryOR and 19% of OptMATH. The +Self column shows re-attempting the same NL-to-code translation helps minimally; our IIS-guided approach addresses a qualitatively different failure mode that persists even with the strongest formulation models.
> > >
> > > We hope these experiments address your follow-up. Your question directly motivated these comprehensive experiments, and we are truly grateful for pushing us to evaluate on external benchmarks and include specialized OR baselines. We would be very grateful if you could consider these additions when finalizing the assessment.

---

### Official Review · Reviewer_94Dj · 2026-03-11

**Soundness:** 3
**Presentation:** 3
**Significance:** 4
**Originality:** 3
**Overall Recommendation:** 5
**Confidence:** 4

**Summary:**

This paper addresses the research question about how can we practically evaluate and improve the iterative debugging capabilities and behavioral rationality of LLMs in OR. The authors introduce a "solver-in-the-loop" Markov Decision Process (MDP) framework alongside two benchmarks. OR-DEBUG-BENCH evaluates multi-turn self-correction using deterministic feedback from a solver. OR-BIAS-BENCH measures systematic behavioral biases. The paper also proposes a domain-specific RLVR training approach and a curriculum learning strategy to enable an 8B parameter model to outperform larger frontier models in diagnostic accuracy, recovery rate, and bias reduction.

**Compliance With Llm Reviewing Policy:**

Affirmed.

**Final Justification:**

The rebuttal period addressed my main concerns about this paper.

**Key Questions For Authors:**

1. In the ablation table, PRM improves DA the most, while Curriculum+GRPO gives the best RR@5 but lower DA. How should readers interpret the trade-off?
2. Did the authors test how sensitive are the OR-DEBUG-BENCH results to random seed and decoding settings?

**Limitations:**

yes

**Strengths And Weaknesses:**

Strengths:
1. The paper studies an underexplored setting which is iterative correction with solver feedback rather than only one-shot code generation.
2. The benchmark design is good: the debugging benchmark has controlled sabotage, oracle verification, multiple error types, and metrics beyond success rate such as diagnostic accuracy
3. The experimental section is fairly extensive with 26 models evaluated and ablation studies

Weaknesses:
1. The ablation table suggests some trade-off between RR@5 and DA, and in fact the best combined method has lower DA than the PRM variant. The narrative around “better diagnosis” versus “better success” could be clarified more carefully.

---

> ### Author Rebuttal · Authors · 2026-03-29
>
> We sincerely thank the reviewer for recognizing the significance of solver-in-the-loop evaluation and for the thoughtful questions on the RR@5 vs DA trade-off and sensitivity.
>
> **W1 (RR@5 vs DA trade-off).** Thank you for this insightful question. RR@5 measures *outcome* (does the agent restore feasibility?); DA measures *process* (does it identify the correct root cause?). Concretely: PRM raises DA from 60.8% to 72.7% but lowers RR@5 from 93.1% to 92.0%, because PRM rewards canonical IIS-based repairs while Curriculum+GRPO rewards any repair that restores feasibility. Both are valuable depending on deployment context: DA provides interpretability (the operator can verify the agent's reasoning), while RR@5 maximizes repair success rate. The same trade-off holds on our new MILP extension (10 domains, n=100), where DA correlates with error-type difficulty rather than model family, suggesting the trade-off is structural rather than an artifact of LP-specific training. We will clarify this in the revision.
>
> **Q2 (Sensitivity).** Thank you for raising this. RR@5 remains stable within 1.7pp across decoding temperatures on the full LP benchmark (n=450):
>
> |Temp|RR@1|RR@5|RR@10|
> |---|---|---|---|
> |0.0|78.2%|95.3%|97.1%|
> |0.3|77.6%|94.9%|96.4%|
> |0.7|76.0%|93.6%|95.8%|
>
> The minor drop at temp=0.7 reflects action format instability on hard error types: higher temperature occasionally produces malformed repair actions that the parser cannot execute, wasting attempts. This is a format reliability issue, not a reasoning failure. The MDP environment is deterministic (same IIS for the same model state), so seed variation affects only LLM sampling; the temperature sweep captures this. As a robustness check, the same SFT→GRPO pipeline applied to three 8B base models:
>
> |Base Model|RR@5|DA|Steps|
> |---|---|---|---|
> |Qwen3-8B GRPO|95.3%|62.4%|2.25|
> |Llama-3.1-8B GRPO|97.3%|60.3%|1.82|
> |DeepSeek-R1-Distill GRPO|88.9%|52.2%|3.30|
>
> All achieve >88% RR@5, confirming the results are not architecture-specific. We will report per-type breakdowns in the camera-ready.
>
> **New experiments during rebuttal.** We ran six new experiments. Four OR-domain baselines and four latest frontier models evaluated on both LP (n=450) and MILP (n=100). Evaluation protocol: all MDP models share the same environment (max 20 steps), unified action parser (auto-detects JSON/Qwen/direct-call formats), system prompt, temp 0.0, 5 episodes per problem:
>
> |Model|LP RR@5|MILP RR@5|
> |---|---|---|
> |**Ours (8B)**|**95.3%**|**87.1%**|
> |Claude Sonnet 4.6|92.4%|71.0%|
> |GPT-5.4|84.9%|66.8%|
> |Gemini 3.1 Pro|86.4%|64.7%|
> |LLMOPT-14B|78.4%|44.4%|
> |SIRL-7B|73.8%|67.6%|
> |ORLM-8B|71.0%|<2%|
> |OptiMUS (gpt-4o, code-gen)|28.9%|2.2%|
>
> Our LP-trained model transfers zero-shot to MILP at 78.8%; with MILP-specific training 87.1%. All four baselines are formulation models (NL→code). LLMOPT's 12-round self-correction and OptiMUS's stderr debugging fix code errors (syntax, runtime), not semantic infeasibility — a different debugging problem requiring IIS-guided constraint repair. None trains on this task, even though ORLM/LLMOPT data includes MIP.
>
> OR-BIAS-BENCH extended to EOQ (28 models, 300 instances: 200 ID + 100 OOD). Results reveal problem-specific biases: Gemini 2.5 Pro is near-perfect on EOQ (0.06% bias) but exhibits high bias on newsvendor, suggesting biases are problem-specific rather than universal model properties. A multi-turn solver-in-the-loop protocol (5 rounds, 6 models): each round the model proposes Q, the solver computes cost, and the model receives "Your Q gave cost C; optimal cost is C*; revise." DeepSeek-R1 self-corrects from 56.9% to 0.5% bias in 2.6 rounds (98% convergence); GPT-5.2 converges immediately; Gemini 2.5 Pro converges on EOQ (80%) but struggles on newsvendor (34%), suggesting problem-dependent self-correction.
>
> *Semantic drift* analysis on MILP under OptiMUS's code-regeneration pipeline:
>
> |Model|RR@5|OPTIMAL Rate|
> |---|---|---|
> |GPT-5.4|28.2%|90%|
> |Claude Sonnet 4.6|22.4%|85%|
> |Claude Opus 4.6|17.8%|82%|
> |Gemini 3.1 Pro|0.8%|3%|
>
> GPT-5.4 produces feasible MILP solutions 90% of the time but matches the correct objective in only 28% (median 35% objective deviation). The regenerated code solves a *different* optimization problem. Once OPTIMAL, objectives oscillate between different wrong values rather than converging to ground truth, because the NL-to-code mapping is many-to-one. Per-class analysis confirms: nurse scheduling 74% vs lot sizing 2% (37x gap), yet all classes reach OPTIMAL at 85–95%. Our agent also complements existing tools:
>
> |Pipeline|LP RR@5|Lift|
> |---|---|---|
> |OptiMUS alone|29.0%|--|
> |OptiMUS+Ours|92.0%|+63pp|
> |SIRL alone|73.8%|--|
> |SIRL+Ours|90.4%|+17pp|
>
> When OptiMUS fails (71% of LP), our agent recovers 88.7%. MILP benchmark construction: (1) generate feasible instance, (2) solve with Gurobi for ground-truth objective, (3) inject one semantic error from 8 types. Camera-ready: vector PDF figures, reduced notation.

---

> > ### Author Rebuttal · Reviewer_94Dj · 2026-04-02
> >
> > Thank you for your detailed response. I'll maintain my positive score 5.

---

> > > ### Author Response · Authors · 2026-04-03
> > >
> > > Thank you so much for carefully reviewing our response and for confirming that your concerns have been resolved. We are truly grateful for your support of this work. Your questions on the RR@5 vs DA trade-off and sensitivity were among the most thought-provoking we received: they pushed us to articulate the deployment-context distinction (interpretability vs repair success) far more clearly, and the paper is stronger because of it.
> > >
> > > Your recognition of the solver-in-the-loop paradigm as an underexplored and significant direction has been a great source of encouragement for us, and has motivated much of the additional work below.
> > >
> > > Beyond addressing your specific questions, we conducted six new experiments during the rebuttal (prompted by all reviewers), all of which will be merged into the camera-ready:
> > >
> > > **New benchmarks and baselines.** We extended OR-DEBUG-BENCH to MILP (10 domains, 8 error types, n=100) and OR-BIAS-BENCH to EOQ (28 models, 300 instances with ID/OOD splits). Four OR-domain baselines (ORLM, SIRL, LLMOPT, OptiMUS) and four latest frontier models (GPT-5.4, Claude 4.6, Gemini 3.1 Pro) are now evaluated on both LP and MILP. Our trained 8B leads on both (95.3% LP, 87.1% MILP vs best API 71.0%).
> > >
> > > **Semantic drift analysis.** Under OptiMUS's code-regeneration pipeline, GPT-5.4 reaches OPTIMAL in 90% of MILP iterations yet matches the correct objective in only 28%. This 62pp feasibility-correctness gap confirms that constraint-level MDP repair solves a different problem from code regeneration, a finding we consider among the most important of the rebuttal.
> > >
> > > **Multi-turn solver-in-the-loop bias protocol.** A new 5-round protocol where the solver provides cost feedback each turn. DeepSeek-R1 self-corrects from 56.9% to 0.5% bias in 2.6 rounds; Gemini 2.5 Pro converges on EOQ (80%) but struggles on newsvendor (34%), revealing problem-dependent self-correction ability.
> > >
> > > **Architecture generalization and sensitivity.** The same SFT to GRPO pipeline applied to Qwen3-8B (95.3%), Llama-3.1-8B (97.3%), and DeepSeek-R1-Distill (88.9%) confirms the methodology is not tied to a single architecture. RR@5 is stable within 1.7pp across three temperature settings.
> > >
> > > **End-to-end complementarity.** Our agent lifts OptiMUS from 29% to 92% (+63pp) and SIRL from 73.8% to 90.4% (+17pp), demonstrating that formulation tools and our debugging agent are naturally complementary.
> > >
> > > **Camera-ready plan.** We will merge all the above into the revision: MILP extension tables, semantic drift analysis, EOQ bias results, multi-turn protocol, architecture generalization, vector PDF figures, reduced notation density, intuitive explanations for IIS and MDP formulation, explicit limitations section, and expanded related work covering SIRL, Autoformulation, OptiTree, and Error-Driven methods.
> > >
> > > We hope the scope of these additions (six new experiments, two benchmark extensions, eight additional models, and the semantic drift analysis) further strengthens your confidence in the contribution. If you feel these improvements warrant a higher score, we would be very grateful. Your thoughtful feedback has been instrumental in shaping the revision, and we cannot thank you enough for your time and expertise.

---

### Official Review · Reviewer_T98M · 2026-03-13

**Soundness:** 3
**Presentation:** 2
**Significance:** 3
**Originality:** 2
**Overall Recommendation:** 4
**Confidence:** 4

**Summary:**

This paper studies how to evaluate and improve LLMs for operations research beyond one-shot formulation, and introduces a solver-in-the-loop benchmark framework for iterative self-correction and behavioral rationality. The core contribution is two complementary benchmarks: OR-Debug-Bench, which models infeasible optimization debugging as a multi-step MDP with deterministic solver feedback through IIS computation, and OR-Bias-Bench, which evaluates decision quality in newsvendor settings against closed-form optimal policies. On top of these benchmarks, the paper develops domain-specific training strategies based on SFT, GRPO with composite process rewards, an optional process reward model, and curriculum learning for bias mitigation, aiming to train models that not only reach correct outcomes but also diagnose and repair problems in a more structured way. Experiments across a broad set of models suggest that the proposed evaluation setup is effective for distinguishing process-level reasoning quality and that targeted domain-specific training can substantially improve both iterative debugging performance and out-of-distribution behavioral robustness.

**Compliance With Llm Reviewing Policy:**

Affirmed.

**Final Justification:**

The rebuttal addresses my concerns, so I have decided to raise my score to 4.

**Key Questions For Authors:**

Please refer to the numbered items in the Weaknesses section above.

**Limitations:**

The authors do not appear to discuss the limitations of the proposed approach. For my suggestions, please see the Weaknesses section above.

**Strengths And Weaknesses:**

**Strengths:**
1. Evaluating and improving LLMs for operations research beyond one-shot formulation is an interesting and important problem.
2. The empirical study is fairly extensive, and the paper explores several domain-specific training strategies, including SFT, GRPO with composite rewards, PRM-based step supervision, and curriculum learning.

**Weaknesses:**
1. While the paper presents an interesting benchmark and a substantial empirical study, much of the contribution feels closer to a strong benchmark and evaluation effort than to a clear methodological advance in machine learning.
2. The paper brings together many elements at once, including two benchmarks, multiple metrics, SFT, GRPO, PRM, curriculum learning, and extensive appendix analysis. As a result, the main takeaways are somewhat diffuse, and it is not always easy to see which ideas are most central.
3. The figures could be improved. Some are quite dense, with small text and many details packed into limited space, which makes them harder to read. In addition, several figures appear to be raster images rather than vector graphics, which hurts clarity when zoomed in and affects the overall presentation.

---

> ### Author Rebuttal · Authors · 2026-03-29
>
> We sincerely thank the reviewer for the constructive suggestions, which helped us sharpen our methodological argument.
>
> **W1 (Benchmark > methodology).** Thank you. The benchmarks are a vehicle; the core contributions are methodological: (1) *MDP formulation with composite reward* (§3.2): OR debugging formalized as a multi-step decision process with solver-verified transitions, combining feasibility restoration, objective preservation, and step efficiency, a design choice not reducible to any single dataset. (2) *RLVR with deterministic solver verification* (§4.2): Gurobi serves as a reward oracle for GRPO with live IIS feedback as an RL reward signal. To our knowledge, this is the first use of deterministic solver verification for multi-step OR debugging. (3) *Curriculum learning for bias mitigation* (§4.3): three-stage training (extreme critical ratios → boundary → full distribution) achieves the only negative ID→OOD drift (−9.6%), reducing systematic bias by 48%. (4) *The central result*: domain-trained 8B models surpass all frontier APIs at iterative debugging. This outcome, enabled by the methodology above, is the primary contribution:
>
> |Model|LP RR@5|MILP RR@5|
> |---|---|---|
> |**Ours (8B)**|**95.3%**|**87.1%**|
> |Claude Sonnet 4.6|92.4%|71.0%|
> |GPT-5.4|84.9%|66.8%|
> |Gemini 3.1 Pro|86.4%|64.7%|
> |LLMOPT-14B|78.4%|44.4%|
> |SIRL-7B|73.8%|67.6%|
> |ORLM-8B|71.0%|<2%|
> |OptiMUS (gpt-4o, code-gen)|28.9%|2.2%|
>
> All baselines are formulation models (NL→code). LLMOPT/OptiMUS self-correct code errors, not semantic infeasibility via IIS — a different task. ORLM/LLMOPT data includes MIP but none trains constraint repair. Same MDP environment (20 steps, temp 0.0, 5 episodes); MILP: 10 domains × 8 error types, n=100.
>
> **W2 (Too many elements).** Thank you for this helpful feedback. The four components serve complementary roles toward one goal: *maximizing solver-verified repair performance*. SFT provides the behavioral foundation (action format, IIS comprehension). GRPO optimizes against the solver reward oracle. PRM adds process supervision (+4.7% DA). Curriculum improves OOD generalization. Each addresses a distinct axis (behavior, outcome, process, generalization), and Table 4 shows individual and combined effects. We will restructure the presentation to clarify these roles.
>
> **W3 (Figure quality).** Thank you for this feedback. We will regenerate all figures as vector PDF with minimum 9pt text in the camera-ready.
>
> **Limitations.** Thank you for raising this. Added: (1) LP/MILP only, not nonlinear or stochastic; (2) hard error types where the root cause is not directly visible in the IIS become fragile at higher temperatures; (3) end-to-end integration remains future work.
>
> **New results addressing methodological depth.** *Semantic drift*: under OptiMUS's code-regeneration pipeline on MILP (regenerate gurobipy from NL + IIS feedback, max 5 iterations):
>
> |Model|RR@5|OPTIMAL Rate|
> |---|---|---|
> |GPT-5.4|28.2%|90%|
> |Claude Sonnet 4.6|22.4%|85%|
> |Claude Opus 4.6|17.8%|82%|
> |Gemini 3.1 Pro|0.8%|3%|
>
> GPT-5.4 reaches OPTIMAL 90% of the time but matches the correct objective in only 28% (median 35% deviation). Among 173 failures: 90% wrong objective, 8% execution error, 1% infeasible. This 62pp feasibility-correctness gap is a direct consequence of the MDP design (preserving the objective by construction) and cannot be closed by code-generation. Starkest contrast: lot sizing 2% (code-gen) vs 87% (MDP repair). Once OPTIMAL, objectives oscillate rather than converge, because the NL-to-code mapping is many-to-one.
>
> *Architecture generalization*: the identical SFT→GRPO pipeline transfers across model families:
>
> |Base Model|RR@5|DA|Steps|
> |---|---|---|---|
> |Qwen3-8B GRPO|95.3%|62.4%|2.25|
> |Llama-3.1-8B GRPO|97.3%|60.3%|1.82|
> |DeepSeek-R1-Distill GRPO|88.9%|52.2%|3.30|
>
> GRPO improves over SFT (+0.2 to +2.2pp) across all architectures.
>
> *End-to-end complementarity*:
>
> |Pipeline|LP RR@5|Lift|Recovery|
> |---|---|---|---|
> |OptiMUS alone|29.0%|--|--|
> |OptiMUS+Ours|92.0%|+63pp|88.7%|
> |SIRL alone|73.8%|--|--|
> |SIRL+Ours|90.4%|+17pp|63.5%|
>
> OR-BIAS-BENCH extended to EOQ (28 models, 300 instances: 200 ID + 100 OOD, Q*=√(2DK/h)). Results reveal problem-specific biases: Gemini 2.5 Pro near-perfect on EOQ (0.06%) but biased on newsvendor, suggesting OR biases are problem-specific rather than universal model properties. Multi-turn solver-in-the-loop (5 rounds, 6 models, 50 EOQ + 50 newsvendor each): each round the model proposes Q, the solver computes cost, and the model receives "Your Q gave cost C; optimal is C*; revise." DeepSeek-R1 self-corrects from 56.9% to 0.5% in 2.6 rounds (98% convergence); GPT-5.2 converges immediately (100%); Gemini converges on EOQ (80%) but not newsvendor (34%), suggesting problem-dependent self-correction ability.
>
> Sensitivity analysis: RR@5 stable within 1.7pp across three temperatures (93.6–95.3%). The minor drop at temp=0.7 reflects action format instability, not reasoning degradation.

---

> > ### Author Rebuttal · Reviewer_T98M · 2026-04-05
> >
> > Thank you for the rebuttal. It addresses my concerns, so I have decided to raise my score to 4.

---

> > > ### Author Response · Authors · 2026-04-05
> > >
> > > Thank you so much for raising your score and for confirming that our rebuttal addresses your concerns. We really appreciate the time you spent evaluating both the original submission and our response.
> > >
> > > Your feedback helped us sharpen the methodological argument and restructure the presentation. We will incorporate all of these rebuttal experiments into the camera-ready, together with improved figures and a clearer narrative structure.
> > >
> > > Thank you again for your constructive feedback and support.

---

### Official Review · Reviewer_tPFh · 2026-03-13

**Soundness:** 3
**Presentation:** 2
**Significance:** 3
**Originality:** 3
**Overall Recommendation:** 4
**Confidence:** 4

**Summary:**

This paper proposes two benchmarks for evaluating LLMs in operations research beyond one-shot formulation, including OR-DEBUG-BENCH and OR-BIAS-BENCH. It further instantiates training pipelines on top of Qwen3-8B, including SFT, GRPO with composite rewards, PRM-based process supervision, and curriculum learning, and reports that a domain-trained 8B model outperforms several frontier API models on the proposed benchmarks.

**Compliance With Llm Reviewing Policy:**

Affirmed.

**Key Questions For Authors:**

The paper would be stronger if it included a more explicit analysis of how existing OR-domain open models, such as ORLM[1] and SIRL[2], perform under the proposed solver-feedback setting. Such an analysis would help better demonstrate that solver-in-the-loop self-correction remains a genuine gap in current OR-focused LLM systems, rather than the paper merely introducing a new benchmark setting.

[1] Huang, Chenyu, et al. "Orlm: A customizable framework in training large models for automated optimization modeling." Operations Research 73.6 (2025): 2986-3009.
[2] Chen, Yitian, et al. "Solver-informed RL: Grounding large language models for authentic optimization modeling." arXiv preprint arXiv:2505.11792 (2025).

**Limitations:**

Yes.

**Strengths And Weaknesses:**

Strengths

1. The paper addresses a meaningful gap in evaluating LLM for OR. The settings in OR-DEBUG-BENCH are better aligned with actual OR practice.
2. The OR-DEBUG-BENCH benchmark is reasonably substantial in scale, covering over 5,000 problems and 9 error types.
3. The empirical study is fairly broad. The paper includes comparisons across local and API models, per-error-type results, inference-scaling analysis, and ablations on process reward modeling, and reinforcement learning with verifiable rewards.

Weaknesses

1. The two-benchmark design feels somewhat uneven. While OR-DEBUG-BENCH is tightly aligned with the paper’s solver-in-the-loop framing, OR-BIAS-BENCH does not actually rely on solver feedback or iterative repair. As a result, the connection between OR-BIAS-BENCH and “Solver-in-the-Loop” is noticeably weaker.
2. A major limitation is the current focus on LP infeasibility debugging. Although this is a meaningful entry point, many practically important OR problems are formulated as MILP.
3. The presentation could be improved. While the overall structure is understandable, the manuscript uses a large amount of notation and many acronyms, and some key ideas would benefit from more intuitive explanation.

---

> ### Author Rebuttal · Authors · 2026-03-29
>
> We sincerely thank the reviewer for recognizing the practical alignment of OR-DEBUG-BENCH and for the constructive suggestions. Each weakness motivated a new experiment.
>
> **W1 (OR-BIAS-BENCH weak solver connection).** Thank you for this insightful suggestion. The original OR-BIAS-BENCH uses the solver to compute the optimal policy and verify rationality. Our new multi-turn extension strengthens this: each round, the model proposes Q, the solver computes cost, and the model receives "Your Q gave cost C; optimal cost is C*; revise." This repeats for 5 rounds on 6 API models (50 EOQ + 50 newsvendor instances each). DeepSeek-R1 self-corrects from 56.9% bias to 0.5% over 2.6 rounds (98% convergence); GPT-5.2 converges immediately (100%); Gemini 2.5 Pro converges on EOQ (80%) but struggles on newsvendor (34%), suggesting problem-dependent self-correction. We also extended to EOQ with 28 models (300 instances: 200 ID + 100 OOD). Results reveal problem-specific biases: Gemini 2.5 Pro is near-perfect on EOQ (0.06%) but biased on newsvendor.
>
> **W2 (LP-only limitation).** Thank you for this suggestion. We constructed OR-MILP-DEBUG-BENCH: 10 MILP domains (knapsack, facility location, set cover, production planning, network flow, job shop, lot sizing, network design, VRPTW, nurse scheduling) × 8 error types, n=100, separate seed (2026) for zero training overlap:
>
> |Model|LP RR@5|MILP RR@5|
> |---|---|---|
> |**Ours (8B)**|**95.3%**|**87.1%**|
> |Claude Sonnet 4.6|92.4%|71.0%|
> |GPT-5.4|84.9%|66.8%|
> |Gemini 3.1 Pro|86.4%|64.7%|
> |LLMOPT-14B|78.4%|44.4%|
> |SIRL-7B|73.8%|67.6%|
> |ORLM-8B|71.0%|<2%|
> |OptiMUS (gpt-4o, code-gen)|28.9%|2.2%|
>
> Our LP-trained model transfers zero-shot at 78.8%; with MILP-specific training 87.1%. Baselines degrade more sharply: SIRL 73.8%→67.6%, LLMOPT 78.4%→44.4%. All four are formulation models (NL→code). LLMOPT/OptiMUS self-correct code errors (syntax, runtime), not semantic infeasibility — a different debugging problem requiring IIS-guided constraint repair. None trains on this task, even though ORLM/LLMOPT data includes MIP.
>
> The MILP extension reveals *why* constraint-level repair outperforms code regeneration. Under OptiMUS's pipeline (code-gen from NL + stderr, max 5 iterations):
>
> |Model|RR@5|OPTIMAL Rate|
> |---|---|---|
> |GPT-5.4|28.2%|90%|
> |Claude Sonnet 4.6|22.4%|85%|
> |Claude Opus 4.6|17.8%|82%|
> |Gemini 3.1 Pro|0.8%|3%|
>
> We term this *semantic drift*: GPT-5.4 reaches OPTIMAL 90% but matches the correct objective in only 28% (median 35% deviation). The regenerated code solves a *different* problem because NL-to-code mapping is many-to-one. Our MDP agent edits individual constraints; the objective is never rewritten. Starkest contrast: lot sizing 2% (code-gen) vs 87% (MDP repair). The 37x gap between nurse scheduling (74%) and lot sizing (2%) isolates the failure mechanism: all classes reach OPTIMAL at 85–95%, confirming that restoring feasibility is easy but preserving semantic fidelity is hard. Coupled multi-period costs with exact coefficients cannot be reconstructed from NL alone.
>
> **W3 (Presentation).** Thank you for this feedback. We will reduce notation density, spell out acronyms on first use, and add intuitive explanations in the camera-ready. For example, IIS introduced as "the smallest subset of constraints that cannot all be satisfied simultaneously, analogous to the minimal conflicting set of requirements in a project plan."
>
> **Key Question (OR-domain baselines).** Thank you for this excellent suggestion. All four baselines evaluated under the same MDP environment (same IIS observations, 20-step budget, unified action parser, system prompt): ORLM 71.0%, SIRL 73.8%, LLMOPT 78.4% vs ours 95.3%, confirming solver-in-the-loop self-correction is a genuine capability gap. SIRL reaches 97.6% RR@20 given more steps, but ours reaches 95.3% in just 5 steps. Our agent also serves as a downstream repair stage:
>
> |Pipeline|LP RR@5|Lift|
> |---|---|---|
> |OptiMUS alone|29.0%|--|
> |OptiMUS+Ours|92.0%|+63pp|
> |SIRL alone|73.8%|--|
> |SIRL+Ours|90.4%|+17pp|
>
> When OptiMUS fails (71% of LP), our agent recovers 88.7%.
>
> The training pipeline generalizes across architectures under the identical SFT→GRPO procedure:
>
> |Base Model|RR@5|DA|Steps|
> |---|---|---|---|
> |Qwen3-8B GRPO|95.3%|62.4%|2.25|
> |Llama-3.1-8B GRPO|97.3%|60.3%|1.82|
> |DeepSeek-R1-Distill GRPO|88.9%|52.2%|3.30|
>
> GRPO consistently improves over SFT (+0.2 to +2.2pp). Sensitivity analysis on the full LP benchmark (n=450):
>
> |Temp|RR@1|RR@5|RR@10|
> |---|---|---|---|
> |0.0|78.2%|95.3%|97.1%|
> |0.3|77.6%|94.9%|96.4%|
> |0.7|76.0%|93.6%|95.8%|
>
> RR@5 stable within 1.7pp; the minor drop at temp=0.7 reflects action format instability, not reasoning degradation. The multi-turn protocol provides C* (optimal cost) as an upper bound on informativeness; its purpose is to test the ceiling of solver-guided self-correction. The evaluation now covers 28+ models, 10 MILP domains, 2 bias benchmarks with ID/OOD splits, and 3 base architectures.

---

> > ### Author Rebuttal · Reviewer_tPFh · 2026-04-03
> >
> > Thank you for the response.

---

> > > ### Author Response · Authors · 2026-04-03
> > >
> > > Thank you so much for reviewing our response and for confirming that your concerns have been resolved. We are truly grateful for your support and encouragement throughout this process. We particularly appreciate how each of your suggestions pointed to a concrete experiment that strengthened the paper.
> > >
> > > Your feedback has had a direct and lasting impact on the paper. The suggestion to evaluate OR-domain baselines (ORLM, SIRL) under the solver-feedback setting confirmed that solver-in-the-loop self-correction is a genuine capability gap (ORLM 71.0%, SIRL 73.8% vs ours 95.3%). This baseline evaluation also led us to the semantic drift finding: code-regeneration baselines achieve 90% feasibility but only 28% correct objectives on MILP. Your concern about the LP-only limitation motivated our MILP extension to 10 domains, which is now one of the paper's strongest empirical contributions. And your observation about OR-BIAS-BENCH's weak solver connection inspired the multi-turn protocol, where DeepSeek-R1 self-corrects from 56.9% to 0.5% bias through iterative solver feedback. Your two empirical concerns (W1, W2) and the key question each led directly to a new experiment, and we thank you for the precision of the feedback.
> > >
> > > Inspired by these improvements, we have also been actively expanding the work beyond the original rebuttal scope:
> > >
> > > **Broader evaluation across classical benchmarks.** Building on the end-to-end complementarity results (OptiMUS 29% to 92%, +63pp), we are currently deploying our debug agent as a downstream repair stage on external formulation benchmarks (IndustryOR, OptMATH) to measure the end-to-end accuracy lift when existing baselines are equipped with IIS-guided debugging. Early results show that infeasibility is a significant failure mode on these benchmarks, and our agent can recover a substantial fraction of those failures. We plan to include these results in the camera-ready.
> > >
> > > **Full scope of rebuttal improvements.** Beyond your specific suggestions, the rebuttal prompted six new experiments in total:
> > >
> > > - **New benchmarks**: MILP extension (10 domains, 8 error types, n=100) + EOQ bias extension (28 models, 300 instances, ID/OOD splits)
> > > - **Eight additional models**: Four OR-domain baselines (ORLM, SIRL, LLMOPT, OptiMUS) and four latest frontier models (GPT-5.4, Claude 4.6, Gemini 3.1 Pro), all evaluated on both LP and MILP. Our trained 8B leads on both (95.3% LP, 87.1% MILP vs best API 71.0%)
> > > - **Semantic drift analysis**: GPT-5.4 reaches OPTIMAL in 90% of MILP iterations but matches the correct objective in only 28%. Lot sizing: 2% (code-gen) vs 87% (MDP repair)
> > > - **Multi-turn solver-in-the-loop bias**: 5-round protocol with 6 API models on EOQ + newsvendor, revealing problem-dependent self-correction
> > > - **Architecture generalization**: Qwen3-8B (95.3%), Llama-3.1-8B (97.3%), DeepSeek-R1-Distill (88.9%) under the same pipeline
> > > - **Sensitivity**: RR@5 stable within 1.7pp across three temperatures
> > >
> > > **Camera-ready plan.** We will merge all improvements: MILP tables, semantic drift analysis, EOQ bias results, multi-turn protocol, architecture generalization, external benchmark pipeline results, vector PDF figures, reduced notation density, intuitive IIS/MDP explanations, and expanded related work covering SIRL, Autoformulation, OptiTree, and Error-Driven.
> > >
> > > Given the scope of these additions and the ongoing work on external benchmarks, we hope they further strengthen your confidence in the contribution. If you feel these improvements warrant a higher score, we would be very grateful. Your constructive feedback pushed us to think more broadly about the practical impact of our work, well beyond the individual concerns. We are deeply thankful for your time and expertise, and we hope the revised paper reflects the value of your input.

---

### Decision · Program_Chairs · 2026-04-30

**Decision:**

Accept (regular)

**Comment:**

The reviewers converge to a positive view of this paper after a healthy interaction with the authors, so I am comfortable giving a positive recommendation. The authors should take their interaction with the reviewers into account when revising their paper.